# NMDA RECEPTOR NONLINEARITY ATTRIBUTES TO MEMORY CONSOLIDATION IN TRANSFORMERS

## ABSTRACT

The NMDA receptor (NMDAR) in the hippocampus is essential for learning and memory. We find an interesting resemblance between deep models' nonlinear activation function and the NMDAR's nonlinear dynamics. In light of a recent study that compared the transformer architecture to the formation of hippocampal memory, this paper presents new findings that NMDAR-like nonlinearity may be essential for consolidating short-term working memory into long-term reference memory. We design a navigation task assessing these two memory functions and show that manipulating the activation function (i.e., mimicking the $Mg^{2+}$-gating of NMDAR) disrupts long-term memory formation. Our experimental data suggest that the concept of place cells and reference memory may reside in the feed-forward network layer of transformers and that nonlinearity plays a key role in these processes. Our findings propose that the transformer architecture and hippocampal spatial representation resemble by sharing the overlapping concept of NMDAR-like nonlinearity.

## 1 INTRODUCTION

In the hippocampus, NMDAR is regarded as an essential component that mediates synaptic plasticity, memory formation, and spatial representation (Li & Tsien, 2009; Tsien et al., 1996; Kentros et al., 1998). NMDAR serves as a switch for synaptic plasticity and long-term memory formation (Bliss & Collingridge, 1993; Slutsky et al., 2010; Miyashita et al., 2012). In addition, NMDAR has been highlighted for its importance in place cell representations in hippocampal CA1 (McHugh et al., 1996; Kentros et al., 1998). Place cells in the hippocampus (O'Keefe & Dostrovsky, 1971) and grid cells in the entorhinal cortex (Hafting et al., 2005) are thought to be crucial for spatial navigation in an animal. These discoveries have triggered recent efforts to replicate these spatial representations found in the brain by using deep neural networks (Banino et al., 2018; Cueva & Wei, 2018; Whittington et al., 2022).

In NMDAR depicted in Fig. 1a, the ion channels that reside in the post-synaptic region have unique characteristics that distinguish them from other ion channels in the brain. Their nonlinear dynamics are modulated by $Mg^{2+}$ ion blockade at the pore region. NMDAR requires activity-dependent repulsion of $Mg^{2+}$ ion (Nowak et al., 1984; Mayer et al., 1984) to be functional, and this phenomenon is partly interesting because it serves as a self-gating of ion influx in the post-synaptic region. In particular, the $Mg^{2+}$ gated nonlinear dynamics of NMDAR plays a key role in synaptic plasticity and memory formation (Slutsky et al., 2010; Miyashita et al., 2012).

Recently, the relationship between the transformer (Vaswani et al., 2017) and hippocampal formation model has been reported (Whittington et al., 2022). The transformer is the most advanced deep learning model, showing unprecedented results in various tasks such as language modeling (Devlin et al., 2018; Brown et al., 2020), computer vision (Dosovitskiy et al., 2020; Radford et al., 2021), and art generation (Ramesh et al., 2022). This model has two consecutive modules, a self-attention layer and a feed-forward network (see Fig. 1b). Whittington et al. (2022) show the self-attention layer is closely related to the state-of-the-art neuroscience model (Whittington et al., 2020) and claim that softmax neurons in the self-attention layer behave like place cells in a navigation task. However, studies on the role of neurons in feed-forward networks have been absent.

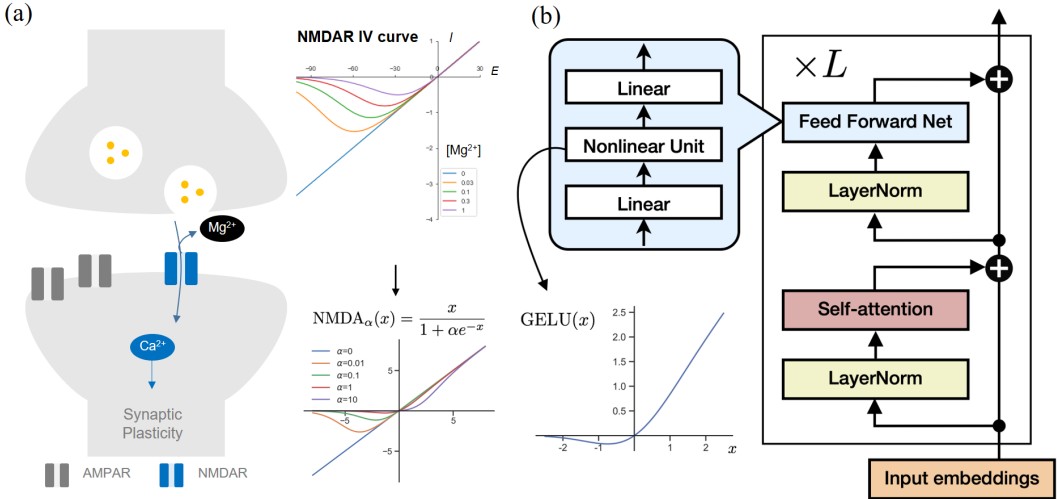

Figure 1: (a) Schematic diagram of $Mg^{2+}$-gated NMDAR modulating synaptic plasticity (left), its IV curve of current-voltage dynamics (right top) and NMDAR-inspired activation function, $\text{NMDA}_\alpha(x)$ (right bottom). (b) Transformer architecture and its feed-forward network's activation function, Gaussian Error Linear Unit (GELU; left bottom).

We find an interesting resemblance of NMDAR nonlinearity with the Gaussian Error Linear Unit (GELU), a nonlinear activation function popularly used in the transformer's feed-forward network (Fig. 1). Similar to NMDAR's activity-dependent gating mechanism of ion influx, the form of the GELU function has a combination of input with the self-gating function. Biological experiments have shown the critical consequence of changing NMDAR's nonlinearity in synaptic plasticity and long-term memory formation (Slutsky et al., 2010; Miyashita et al., 2012), while the role of NMDAR-like nonlinearity in place cell representation remains unclear.

This work is inspired by the fascinating resemblance of NMDAR's nonlinearity dynamics with the GELU activation function and the recent model relating transformer's self-attention mechanism to hippocampal formation (Whittington et al., 2020; 2022). These findings motivated us to ask a question; **Can NMDAR-like nonlinearity in the feed-forward network layer of transformers enhance the formation of long-term memory and spatial place cell representation?**

To address this question, we propose a novel NMDAR-like activation function derived from NMDAR IV curve and design a spatial navigation task in a 2D grid environment that can assess two different memory types well formulated in neuroscience experiments (Olton et al., 1977; 1979): working memory and reference memory. Working memory controls the events from a within-trial, while reference memory controls across-trials from the unchanging environment. We evaluate the transformer model with the NMDAR-like activation function on this task; the results show that 1) place cell representations emerge in feed-forward networks, 2) the reference memory can be controlled by the nonlinearity of the NMDAR-like activation function, 3) the place cell in feed-forward networks is strongly correlated with the reference memory, while the place cell in self-attention layers has no correlation, 4) the proposed NMDAR-like activation shows the best reference memory performance when compared to other widely used nonlinear activation functions.

Our experimental data suggest that NMDAR-like nonlinearity in the feed-forward network layer of the transformer can enhance the long-term memory formation and place cell representation.

## 2 TRANSFORMER

The transformer architecture (Vaswani et al., 2017) can be constructed by stacking multiple blocks of self-attention layers and feed-forward networks (see Fig. 1b). Here we briefly review the self-attention mechanism and the feed-forward networks.

**Self-attention mechanism**  Given a sequence $\{\mathbf{x}_1, ..., \mathbf{x}_T\}$ of $d$-dimensional input embeddings, the self-attention layer calculates the interaction term between each embedding element within a context window via the self-attention mechanism. More formally, each input embedding applies two linear layers ($W_k$ and $W_v$) to the embeddings to form the key matrix $K$ and value matrix $V$:

$$K^\top = [\mathbf{k}_{t-c}^\top \ \mathbf{k}_{t-c+1}^\top \ ... \ \mathbf{k}_t^\top], \quad \text{where } \mathbf{k}_i = \mathbf{x}_i W_k \ (W_k \in \mathbb{R}^{d \times d_k});$$
$$V^\top = [\mathbf{v}_{t-c}^\top \ \mathbf{v}_{t-c+1}^\top \ ... \ \mathbf{v}_t^\top], \quad \text{where } \mathbf{v}_i = \mathbf{x}_i W_v \ (W_v \in \mathbb{R}^{d \times d_k}). \tag{1}$$

Here, $c$ denotes the context length. The key matrix $K \in \mathbb{R}^{(c+1) \times d_k}$ is then used to compute the interaction score between an input embedding at step $t$ and all the vectors in $K$ via dot products:

$$\mathbf{s}_t = \mathbf{q}_t K^\top, \quad \text{where } \mathbf{q}_t = \mathbf{x}_t W_q \ (W_q \in \mathbb{R}^{d \times d_k}). \tag{2}$$

The normalized values of $\mathbf{s}_t \in \mathbb{R}^{(c+1)}$, called attention values, are calculated via the softmax function; the final output of the self-attention mechanism is a weighted sum of the value vectors in $V \in \mathbb{R}^{(c+1) \times d_k}$ with the attention values:

$$\mathbf{y}_t = \texttt{softmax}\left(\frac{\mathbf{q}_t K^\top}{\sqrt{d_k}}\right) V. \tag{3}$$

After this update, $\mathbf{y}_t \in \mathbb{R}^{d_k}$ is updated by another linear transformation $W_o \in \mathbb{R}^{d_k \times d}$: $\mathbf{z}_t = \mathbf{y}_t W_o$. The output $\mathbf{z}_t$ is added to the $\mathbf{x}_t$; $\mathbf{z}_t + \mathbf{x}_t$ is the final output of the self-attention layer, and this information is sent through the following subsequent layer.

**Feed-forward networks**  Another important component of a transformer layer is the feed-forward network. This consists of two linear layers with a point-wise nonlinear activation function $\phi$:

$$\text{FFN}(\mathbf{x}_t) = \phi(\mathbf{x}_t U_1^\top) U_2, \tag{4}$$

where $U_1 \in \mathbb{R}^{d_f \times d}$ and $U_2 \in \mathbb{R}^{d_f \times d}$ are trainable weight matrices. Sukhbaatar et al. (2019) pointed out that equation 4 and equation 3 have similar structures except for the following two major differences: 1) $U_1$ and $U_2$ matrices are fixed over different input sequences while $K$ and $V$ matrices are dynamically changed as input is and 2) operations in feed-forward networks are entirely point-wise or local while the self-attention layer has non-local operations, e.g., the softmax function and dot products between different elements. This observation suggests that the feed-forward networks store general knowledge about the task that does not depend on the situation.

## 3 METHODS

### 3.1 RELATING ACTIVATION FUNCTION IN TRANSFORMERS WITH NMDAR NONLINEARITIES

NMDAR's nonlinear dynamics arise from the voltage-gated $Mg^{2+}$ repulsion at the NMDAR channel's pore (Nowak et al., 1984; Mayer et al., 1984) (Fig. 1a left). Previously, $Mg^{2+}$-gated NMDAR open probability $\mathbf{p}$ has been shown to follow ion blockade model of Woodhull (1973):

$$\mathbf{p}_\alpha(x) = \frac{1}{1 + \alpha e^{-\beta x}}, \tag{5}$$

where $x$ represent an input voltage, $\alpha = [Mg^{2+}]/K_{Mg^{2+}}$ is a parameter determined by $[Mg^{2+}]$, $K_{Mg^{2+}}$ is a dissociation constant, and $\beta$ is a temperature constant. Experimentally, increasing the $Mg^{2+}$ level in the brain can enhance long-term memory formation (Slutsky et al., 2010). We observed the NMDAR's nonlinear dynamics of $IV$ curve (Fig. 1a right top; current-voltage relationship) in the synapse to closely resemble the form of GELU activation function. GELU is a widely used activation function in transformers (Fig. 1b left bottom; $\text{GELU}(x) \approx x\sigma(1.702x)$ where $\sigma$ is the sigmoid function) (Hendrycks & Gimpel, 2016; Devlin et al., 2018; Brown et al., 2020). Inspired by this resemblance, we define a new nonlinear activation function (Fig. 1a right bottom) with $\alpha$ parameter which modulates dynamics as following (see details in Appendix A.1 & A.2):

$$\text{NMDA}_\alpha(x) = x\mathbf{p}_\alpha(x) = \frac{x}{1 + \alpha e^{-x}}. \tag{6}$$

To investigate this NMDAR-like nonlinearity in transformer memory formation, we replaced the $\text{GELU}(x)$ activation function with $\text{NMDA}_\alpha(x)$ in a standard transformer model.

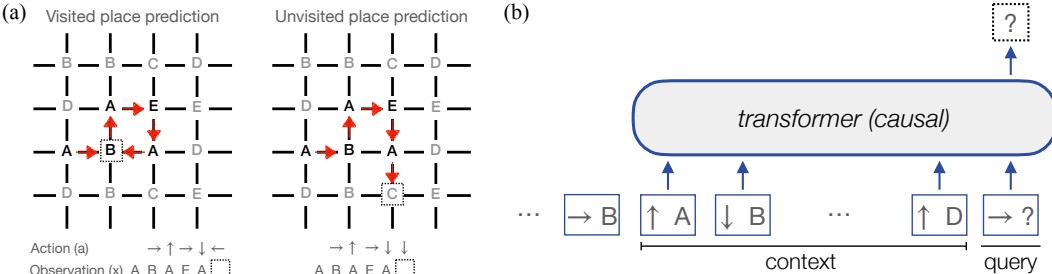

Figure 2: (a) Sensory observation prediction task in a 2D grid, where dotted squares indicate the target position to predict given a sequence of past actions and observations. The unvisited (visited) places are represented in gray (black) letters. (b) A transformer model for predicting the next location's observation based on sequences of [action, observation] pairs. Using the sequence of pairs in the context, the model is trained to predict the masked observation (i.e., the subsequent observation) corresponding to the final query action.

## 3.2 TRANSFORMERS LEARN SPATIAL NAVIGATION TASKS

We train the transformer model to predict the subsequent sensory observations of an agent that randomly walks a 2D grid environment (Whittington et al., 2022) (Fig. 2a). A sequence of previous [Action ($a$), Observation ($x$)] pairs is an input to the model, and the subsequent observation is masked for prediction (Fig. 2b). Instead of using sinusoidal positional encoding (Vaswani et al., 2017) that is commonly used in transformers, we employ the recurrent positional embedding which is encoding the location of an input element by using the recurrent neural network (RNN) (Whittington et al., 2022)[1].

We generate the embedding vectors of the sensory observation sequence with a word embedding layer, but the embedding vectors of the action sequence are generated by RNN; $\mathbf{e}_{t+1} = \tanh(\mathbf{e}_t W_a)$, where $\mathbf{e}_t$ is a recurrent positional embedding at step $t$, and $W_a$ is the action-dependent trainable weight matrix. The input is given by $\{[\mathbf{e}_1, \mathbf{x}_1], [\mathbf{e}_2, \mathbf{x}_2], \ldots, [\mathbf{e}_t, \mathbf{x}_t]\}$, where $\mathbf{x}$ denotes the embedding vector of sensory observation $x$; the initial recurrent positional embedding $\mathbf{e}_1$ is sampled from a normal distribution and we mask the last observation $x_t$. We generate $N$ maps of $11\times 11$ 2D grids. A random sensory observation among ten letters is placed at each position on each map. Agents can move 'up,' 'right,' 'down,' 'left,' or 'stay.' An agent starts at a random position and initiates a random walk on the map, a randomly selected map among $N$ training maps, for 2,048 steps for each trial.

The model is trained with the softmax cross-entropy loss and predicts the subsequent sensory observations (i.e., dotted squares). We evaluate two types of memory: **working memory** and **reference memory**. When the prediction on nodes that were previously visited during the random walking is incorrect, it will count as the working memory error (see Fig. 2a left). On the other hand, when the prediction on unvisited nodes is incorrect, it will count as the reference memory error (see Fig. 2a right). Minimizing the reference memory error by memorizing input sequences is infeasible; the possible number of sequence configurations is exponential since the input sequence is randomly generated at each trial. To solve this task, the model should be able to 1) understand the abstract structure of 2D space, 2) infer which map it is on from input sequence data, and 3) memorize what sensory observation is placed at each position in that map. We describe more training, evaluation, and transformer model details in the following section. See details for task and definition regarding working and reference memory in Appendix A.3.

## 4 RESULTS

### 4.1 IMPLEMENTATION DETAILS

In our experiment, the feed-forward network (FFN) in the transformer model consists of two linear layers (see Fig. 1b and equation 4) with the NMDAR-inspired activation function NMDA$_\alpha$ (Eq. 6).

---

[1]This method is closely related to the most advanced neuroscience model of the hippocampus.

We use TransformerXL (Dai et al., 2019) with an extended memory length of 32 and segment length of 32 so that the context length $c$ is 64 and working memory error is measured when the node to predict its sensory observation is in the context window (see Fig. 2b); i.e. a node that the agent had never visited within recent 64 steps is treated as an unvisited node. The input embedding is concatenated vector $[\mathbf{e}, \mathbf{x}]$ of the word embedding $\mathbf{x}$ (dimension of 256) and the recurrent positional embedding $\mathbf{e}$ (dimension of 256) so that the total input embedding dimension is 512. The number of heads in the self-attention layer is 8 and the number of neurons in the FFN is 2,048. The dropout rate is set to 0.1 and the maximum clip norm of the gradient is set to 0.25. We employ ADAM (Kingma & Ba, 2015) optimizer and a learning rate schedule with a linear decay from 0.0001 (start) to 0 (end). We run 512 random walk simulations (trials) in parallel for collecting training trajectories. The total number of random walking steps is 2,048 for each simulation so the total number of steps for training a model is 512 (batch size; number of trials per epoch) × 2,048 (total number of steps in a trial) × 200 (number of epochs) (see Fig. 7 in Appendix A.3). All runs are performed on a single NVIDIA TITAN V GPU.

## 4.2 WORKING MEMORY ERROR & REFERENCE MEMORY ERROR

To measure the impact of nonlinearity $\alpha$ in the FNNs, we train the transformer models with different values of $\alpha$ in $[0, 0.01, 0.05, 0.1, 0.5, 1, 5, 10]$ and evaluate the working memory and reference memory errors on the train maps (i.e., familiar maps) and test maps (i.e., novel maps). The average number of unvisited nodes in a single trial is 561.

The top left plot in Fig. 3a shows that the reference memory error on the train maps is rapidly decreased over train trials when $\alpha$ is larger than zero, with a larger improvement for increasing $\alpha$. The reference memory error on the novel maps, however, is nearly constant at the chance level of $0.9 (= 1 - 1/(\text{number of letters}))$ for all $\alpha$ (see Fig. 3a top right). Fig. 3a (bottom right) shows that working memory is active on novel maps that had not been shown during training. This finding suggests that the working memory formation is intact on novel maps. Training the models on different numbers of maps $N$, Fig. 3b shows that increasing nonlinearity (i.e., $\alpha$) helps activate the reference memory, and the trend of improvement is consistently shown for $N = 32, 48,$ and $64$ cases. Training over more maps leads to bigger reference memory errors. This is because more

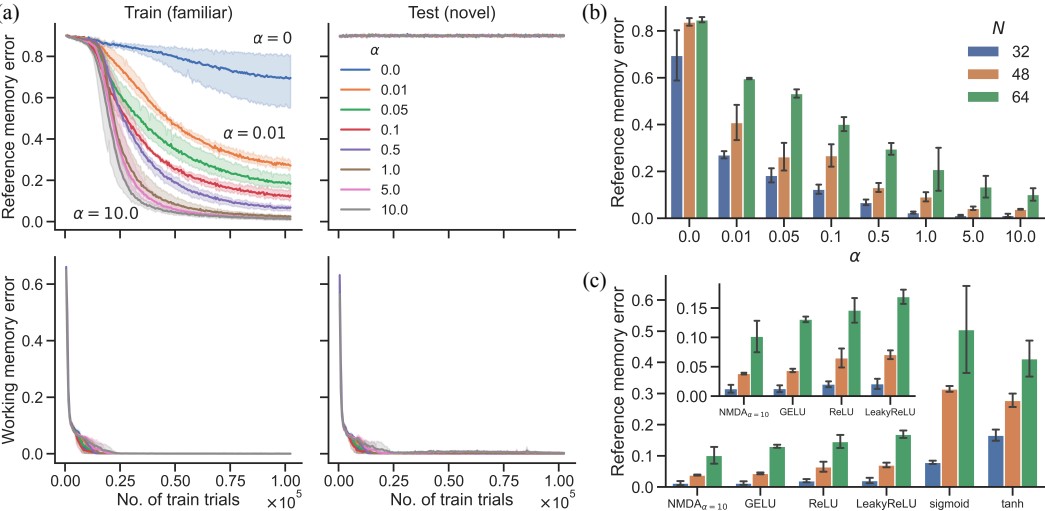

Figure 3: (a) Reference and working memory errors over training trials for training (familiar) maps and testing (novel) maps for $N = 64$ where $N$ is the number of training maps. (b) Reference memory errors evaluated on training maps over different values of $\alpha$ in $\text{NMDA}_\alpha$ and $N$. (c) Reference memory errors comparison between $\text{NMDA}_\alpha = 10$, GELU, ReLU, LeakyReLU, sigmoid, and tanh activation functions. Inset: zoom on the top 4 activation functions. Error bars and shaded areas represent the standard deviation of errors from three independently trained models.

maps require the model to store more pairs of 'what'-'where' memory (i.e., each training contains unique 'what'-'where' information).

In addition, we demonstrate other nonlinear activation functions which are widely used in the machine learning literature. We test GELU ($x\sigma(1.702x)$), ReLU ($\max(0, x)$), LeakyReLU ($\max(0, x)+0.01 \min(0, x)$), sigmoid, and tanh in the FFNs. As can be seen in Fig. 3c, NMDA$_{\alpha=10}$ shows the lowest reference memory errors on the training maps.

Other memory types, such as information in path integration (i.e., recurrent positional embedding), may be used instead of reference memory. To test this assumption, we used non-recurrent positional embeddings to train the models. The result shows that working memory and reference memory errors increase substantially. However, it exhibits similar behavior to the trend of decreasing reference memory error while increasing $\alpha$ of NMDA$_\alpha$ (see Fig. 8 in Appendix A.4). We also assessed the prediction error of the first visited node. While the reference memory error is defined as a prediction error on a node that the agent has not visited in the previous 65 steps, the first visited node prediction error is a prediction error on a node that the agent visits for the first time in a trial. The results for the first visited node prediction error in training maps are identical to the results for the reference memory error (see Fig. 9 in Appendix A.4). These findings suggest that the reference memory is used in training maps to predict the unvisited node.

### 4.3 PLACE CELLS IN FEED-FORWARD NETWORKS

Place cell is a neuron in the hippocampus which fires at a particular place of the environment (O'Keefe & Dostrovsky, 1971). Studies have shown that hippocampal place cells encode the spatial location through localized firing patterns. They have been considered a substrate for long-term memory of the location where specific events occurred (i.e., previously visited position in our navigation task). Selective impairment of NMDAR in hippocampal CA1 disrupts place cell emergence and long-term memory formation (Tsien et al., 1996; Kentros et al., 1998; McHugh et al., 1996).

We investigate the role of neurons in the FFNs and self-attention layers by measuring the neuron's place specificity. Given a $K \times K$ 2D grid environment as graph $G = (V, E)$ and a firing rate

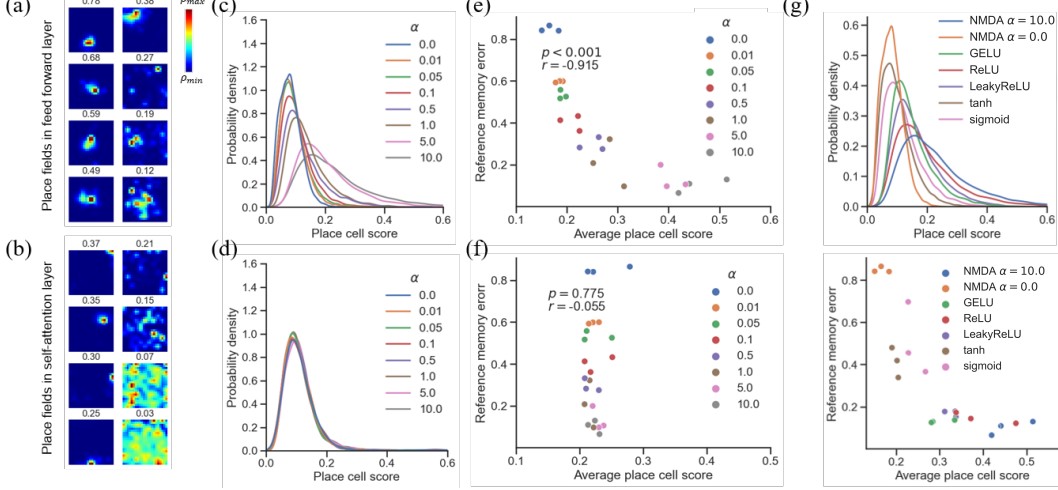

Figure 4: Reference memory-related place cells selectively emerge in the feed-forward layer but not in the self-attention layer along with $\alpha$ increase ($N = 64$). (a, b) Example rate maps with place scores in feed-forward layers and self-attention layers at $\alpha = 10$; from top left (high) to bottom right (low); color bar indicates the firing rate between $\rho_{max}$ and $\rho_{min}$. (c-d) Place cell score distributions with varying $\alpha$ in feed-forward layers (c) and self-attention layers (d). (e-f) Scatter plot of average place cell scores and reference memory errors. $r$ and $p$ denote Spearman's rank correlation coefficient and significance score, respectively. (g) place cell score distribution and relationship of average place cell scores and reference memory errors in common activation functions: GELU, ReLU, LeakyReLU, tanh, and sigmoid. All results are evaluated from training maps.

(cumulative activation value at node $i$ divided by the length of evaluation trial) of node $i \in V$ as a $\rho_i$, we define maximally firing node as $i_{\max}$ and its firing rate as $\rho_{\max}$. Where $E$ is directed edges, which connect high to low firing nodes in $G$. From $G$, we run depth-first-search from source node, $i_{\max}$, to build a sub-graph $\mathcal{G} = (\mathcal{V}, \mathcal{E})$ which we call all connected components. Given $G$ and $\mathcal{G}$, the place cell score is defined as following

$$\text{Place cell score} = \gamma \frac{\sum_{i \in \mathcal{V}} \rho_i}{\sum_{i \in V} \rho_i}, \tag{7}$$

where $\gamma = 1 - |\mathcal{V}^*|/|V|$ is a discount factor and $\mathcal{V}^*$ is $\mathcal{V}$ without node $i_{\max}$ and leaf nodes (see details in Appendix A.6). To measure place cell score, we record the firing rate $\rho_i$ of neurons over a random walking trajectory with $10^5$ steps in one of the training maps; then we measure the place cell scores of neurons in the FFN and self-attention layers. The place cell score is 1 when the neuron is firing only at a certain node; the score is 0 when the neuron is firing homogeneously across all nodes.

Fig. 4a and 4b show the rate maps of neurons with place cell scores in the FFN and self-attention layers, respectively. For the self-attention layer, the total number of neurons in the softmax layer is 65 (context length + masked sensory observation) $\times$ 8 (number of heads) $\times$ 2 (number of layers). The total number of neurons in the FFN layer is set as 2,048 (number of neurons) $\times$ 2 (number of layers). As can be seen, our metric well represents place specificity. Fig. 4c and 4d show the distribution of place cell scores in the two layers with different values of $\alpha$. When the $\alpha$ value is increased, the place cell score distribution found in the FFN layer becomes positively shifted (see Fig. 5 rate map examples for $\alpha = 0$, 1.0, and 10.0), whereas place cell score distribution in the self-attention layers remains.

Fig. 4e and 4f show a relationship between the average place cell score and the reference memory error for each $\alpha$. While average place cell scores in self-attention layers show no correlation with reference memory errors whatsoever, neurons in the FFN exhibit a substantial correlation. These results imply that the reference memory formation and place cell emergence can be enhanced by NMDAR-like nonlinearity in the FFN.

In Fig. 4g, we compare the place cell representations of our NMDA ($\alpha = 0, 10$) with the representations in FFNs with the activation functions used in Fig. 3c. Our results show that the case of NMDA$_{\alpha=10}$ outperforms other activation functions, in both reference memory formation and place cell representation. Our finding that increasing $\alpha$ ([Mg$^{2+}$] component) enhances reference memory is in line with the biological observation that increasing the [Mg$^{2+}$] in the brain enhances long-term memory formation (Miyashita et al., 2012).

In addition, we investigate the consequence of changing nonlinearity with other than NMDA$_\alpha$. We choose LeakyReLU with controllable negative slope ($\max(0, x) + \alpha \min(0, x)$) to compare with NMDA$_\alpha$. Compared to NMDA$_{\alpha=10}$, LeakyReLU exhibits a lower average place score in the allowed range of $\alpha$, indicating that NMDA$_\alpha$ is better in place cell emergence (see Fig. 10 in Appendix A.7).

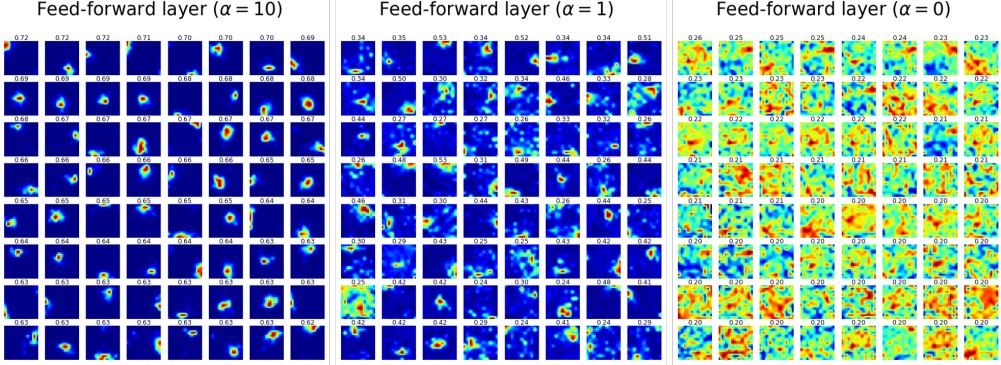

Figure 5: Rate maps of neurons with top-64 place cell scores in the feed-forward network with varying values of $\alpha$; $\alpha = 10$ (left), $\alpha = 1$ (middle), and $\alpha = 0$ (right).

## 5 RELATED WORKS

The current study is inspired by recent observations that connect neuroscience and AI models. One such seminal work is by Whittington et al. (2022), where the authors showed the relationship between the self-attention layer and the state-of-the-art hippocampal model called the Tolman-Eichenbaum Machine (TEM; Whittington et al. (2020)). The current work expands the literature by focusing on the feed-forward networks (FFNs) in the transformer and the emergence of place cells.

TEM is a neuroscience-based model that reproduces neural representations in the hippocampus and entorhinal cortex. Instead of storing memory in the key matrix $K$ and value matrix $V$, it assumes to store memory at the Hebbian weight matrix $M \in \mathbb{R}^{d_k \times d_k}$ and every outer product of key and value vector $\mathbf{k}_i^\top \mathbf{v}_i$ at each step $i$ are simply stored in $M$ via Hebbian update rule. $M$ is initialized to a zero matrix at the beginning of the task and adds every outer product at each time step:

$$M = a \sum_{i=1}^{t} \mathbf{k}_i^\top \mathbf{v}_i = a K^\top V, \tag{8}$$

where $a$ is a weighting factor. In the memory retrieving phase with the query vector $\mathbf{q}$, TEM uses an attractor network[2]:

$$\mathbf{q}M = a\mathbf{q}K^\top V. \tag{9}$$

Whittington et al. (2022) found that the memory retrieving process in TEM has a close mathematical structure with equation 3 when the softmax function is replaced with a linear function. Their subsequent model, called TEM-t (Whittington et al., 2022), replaces the attractor network (equation 9) with self-attention mechanism (equation 3). They demonstrated that TEM-t learns significantly faster than TEM.

TEM-t and TEM do not have a fixed context length $c$; therefore, these models store all information before step $t$, i.e., $c = t$. The computational cost of the self-attention layer in TEM-t is $O(t^2)$, and retaining all previous information is too expensive from both biological and computational standpoint[3]. For TEM, the Hebbian update rule has no quadratic computational cost and can add all previous information in a fixed number of synapse $d_k^2$; however, the memory capacity of the Hebbian matrix $M$ is $O(d_k)$ and the speed of memory retrieval is substantially slower than the self-attention mechanism (Demircigil et al., 2017; Ramsauer et al., 2021; Krotov & Hopfield, 2021). In contrast to TEM and TEM-t that rely on a single memory system, the transformer model employs two separate memory systems: 1) context-dependent matrices $K$ and $V$ in the self-attention layer with a fixed context length $c$ and 2) context-independent fixed matrices $U_1$ and $U_2$ (in equation 4) in the FFNs.

We also focus on the observation by Whittington et al. (2022) that softmax neurons in the self-attention layer behave like place cells. Nonetheless, the role of neurons in FFNs of transformers has not been thoroughly investigated, which is our contribution. In our work, we newly 1) propose a method for assessing the reference memory; 2) compare the effects of various nonlinear activation functions in FFNs with our NMDA-inspired activation functions on reference memory performance; 3) demonstrate the emergence of place cells in FFNs. We note that TEM and TEM-t only evaluated working memory errors in test maps.

## 6 DISCUSSION AND CONCLUSION

**Searching for biological substrate of nonlinear activation function**  Rigorous previous efforts in finding the optimal nonlinear activation function underlie the great success of modern deep neural network models (Nair & Hinton, 2010; Hendrycks & Gimpel, 2016; Ramachandran et al., 2017). However, the neural substrates that mediate nonlinearity in the human brain and their role in intelligence have not been clearly understood. Our work is one of the first to put together the biologically inspired nonlinearity and its effect on long-term memory formation and the place cell representation in the previously described transformer model of the hippocampal formation. This idea was tested on a sensory observation task in the 2D grid environment and with the implementation of NMDAR-like nonlinearity. Our data indicated that NMDAR-like nonlinearity in the feed-forward network

---

[2]Note that this is a simplified description of TEM and it is not exactly the same.

[3]Due to this limitation, TEM-t does not store all historical data. Instead, the model selectively chooses which data to store in $K$ and $V$.

layer of transformers can enhance the formation of long-term memory and spatial place cell representation. Furthermore, this design choice improves long-term memory more than other commonly used nonlinear functions.

Our results agree qualitatively with previous NMDAR impairment experiments from neuroscience: 1) hippocampal CA1 NMDAR perturbation does not impair working memory (Lee & Kesner, 2002), 2) changing NMDAR $Mg^{2+}$-gating (changing $\alpha$ in this work) enhances or disrupts long-term memory formation (Slutsky et al., 2010; Miyashita et al., 2012), 3) NMDAR is required for long-term stabilization of newly forming place fields (McHugh et al., 1996; Kentros et al., 1998). Our contribution is at showing these patterns experimentally for the first time.

**Short-term working memory and long-term reference memory**  Memories can be divided into short-term memory and long-term memory with respect to time (Atkinson & Shiffrin, 1968). Later, these two different memory systems were denoted as working memory and reference memory, concerning functional aspect (Baddeley & Hitch, 1974). In neuroscience, there is a consolidation theory that some short-term working memories are converted to a long-term reference memory system, and others fade out. There has been accumulating evidence that NMDAR is essential for memory consolidation in the hippocampal CA1 region (Kentros et al., 1998; Shimizu et al., 2000). Our work assessed the short-term working memory and long-term reference memory in the transformer's navigation task by measuring the visited error and unvisited error, respectively.

The modulation of $\alpha$ selectively affects the formation of long-term reference memory (i.e., prediction of unvisited places across trials) while leaving the formation of short-term working memory (i.e., prediction of unvisited places within trials) intact. This result suggests that short-term working memory and long-term reference memory are physically stored in separate structures: the self-attention layer and the feed-forward layer. A similar idea has been proposed in psychology, which we illustrate in detail in Appendix A.5.

In neuroscience, the transfer of short-term memory into a long-term system is called *memory consolidation* (McGaugh, 2000). Various research has revealed that $Mg^{2+}$-gating of NMDA receptors modulates the formation of long-term memory (Slutsky et al., 2010; Miyashita et al., 2012). These observations imply that the nonlinear dynamics of NMDA receptors in hippocampus CA1 are critical for consolidating short-term memory into long-term memory.

Current research showed that the transformer could model memory consolidation. We assumed that the GELU activation function links short-term working memory and long-term reference memory. Our experiments indicate that the formation of long-term reference memory is impaired when the activation function is completely linear (corresponding to no $Mg^{2+}$). In contrast, increasing $\alpha$ (which corresponds to an increase in $Mg^{2+}$ level) has resulted in superior performance in long-term reference memory compared to other activation functions (e.g., ReLU, GELU, LeakyReLU, sigmoid, tanh). These similarities between hippocampal memory consolidation and our results suggest a transformer as an effective memory consolidation model.

We have investigated the role of NMDAR-like nonlinearity with a novel consideration of activation function in a transformer model and have demonstrated the effects of altering the nonlinear dynamics of the activation function. Even though there are trainable parameters in the self-attention layer, the quantitative analysis of the place cell score indicates that most of the reference memory is stored in feed-forward networks. Surprisingly, our result that loss of nonlinearity in NMDAR-like activation function ($\alpha = 1$) show impaired reference memory formation (Fig. 3b) but intact working memory formation (Fig. 3a) seems to replicate the previous findings in neuroscience; selective inhibition of hippocampal CA1 NMDAR inhibition does not disrupt working memory (Lee & Kesner, 2002) but impairs the long-term memory formation (Tsien et al., 1996). These similarities provide an exciting possibility that our brain selectively updates short-term working memory into long-term reference memory by activity-dependent $Mg^{2+}$-gating of NMDAR.

**Future directions**  Our research has exciting future directions. The current study only examined what-where memory using a sensory observation task in a static environment. However, our real-world environment is changing dynamically. Modern deep learning systems are generally incapable of adapting to a dynamic environment or reordering of sensory inputs. In future work, we intend to explore what-where-when memory, called *episodic memory*, in transformer and other deep models.

## 7 REPRODUCIBILITY STATEMENT

We included the Pytorch implementation code available along with the paper (see the Supplemental Material file). For reproducibility, we included a thorough overview of the experimental setup, task design, and place cell score evaluation pseudo-code in the Appendix.

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

# A APPENDIX

## A.1 DERIVATION OF NMDAR NONLINEARITY FROM THE MOLECULAR LEVEL CHEMICAL INTERACTION

Here, we describe the NMDAR nonlinear dynamics from chemical interaction between $Mg^{2+}$ and NMDAR following previous literature Woodhull (1973); Jahr & Stevens (1990); Perouansky & Yaari (1993). At the molecular level, one $Mg^{2+}$ ion binds to one NMDAR receptor when opening the NMDAR channel. Thus, the chemical equation of binding reaction between $Mg^{2+}$ ion and NMDAR receptor, R, can be described as

$$Mg^{2+} + R \rightleftharpoons Mg^{2+}R. \tag{10}$$

From this chemical equation, the equilibrium constant $K$ is given by

$$K = \frac{[Mg^{2+}R]}{[Mg^{2+}][R]}. \tag{11}$$

Thus, dissociation constant $K_D$, which correspond to $Mg^{2+}$ dissociation from NMDAR, follows

$$K_D = K^{-1} = \frac{[Mg^{2+}][R]}{[Mg^{2+}R]}, \tag{12}$$

in which [R] and $[Mg^{2+}R]$ are the free and $Mg^{2+}$-bound NMDARs respectively. The fraction of opened NMDAR channels (number of unbound NMDAR over a number of total NMDAR) at equilibrium follows,

$$\begin{aligned} \mathbf{p} &= \frac{[R]}{[R] + [Mg^{2+}R]} \\ &= \frac{1}{1 + [Mg^{2+}]/K_D} \end{aligned} \tag{13}$$

Experimentally, the voltage-dependent dynamics of $K_D$ has been described as following equation by Ascher & Nowak (1988)

$$K_D = K_{Mg^{2+}} e^{\beta V}, \tag{14}$$

where, $V$ is membrane voltage, $\beta$ is a temperature constant and $K_{Mg^{2+}}$ is a dissociation constant at $V = 0$. If Eq.14 is substituted into Eq.13, voltage-dependent open fraction of NMDAR can be expressed as follows

$$\begin{aligned} \mathbf{p}(V) &= \frac{1}{1 + \frac{[Mg^{2+}]}{K_{Mg^{2+}}} e^{-\beta V}}. \\ &= \frac{1}{1 + \alpha e^{-\beta V}}. \end{aligned} \tag{15}$$

in which $\alpha = [Mg^{2+}]/K_{Mg^{2+}}$, the parameter determined by the $[Mg^{2+}]$. Given the voltage-dependent open fraction of NMDAR, $\mathbf{p}(V)$, and NMDAR's maximal conductance, $g_{max}$, then voltage-dependent NMDAR conductance $g(V)$ can be described as

$$g(V) = g_{max}\mathbf{p}(V) \tag{16}$$

Given $g(V)$, and driving force, $V - V_r$, and current $I$, they have a relationship of $I = (V - V_r)g(V)$, in which $V_r$ is reversal potential (the value of membrane potential above which current inverts the direction). As experimental investigations on the physiological reversal potential of NMDAR to be

$V_r = 0$ Mayer & Westbrook (1987); Ichinose et al. (2003); Kohr et al. (1993), $I = Vg(V)$. Then, the normalized NMDAR current $I_{norm} = I/g_{max}$ follows

$$I_{norm} = V\mathbf{p}(V) \tag{17}$$

From Eq. 17 and previous electrophysiological experimental results Kirson et al. (1999), we reconstruct IV curve in (Fig. 1a, right top).

## A.2  NMDAR-INSPIRED NONLINEAR ACTIVATION FUNCTION

Here, we propose an NMDAR-inspired nonlinear activation function from the nonlinear dynamics of the NMDAR-IV curve. If we consider the nonlinear IV curve of NMDAR (Eq. 17) as a nonlinear mapping function, $\phi$, the membrane voltage, $V$, can be viewed as an input, $x$, and normalized NMDAR current, $I_{norm}$, as an output, $\phi(x)$. Then we can rewrite the nonlinear mapping function $\phi$ as follows

$$\phi(x) = x\mathbf{p}(x). \tag{18}$$

We define the NMDAR-inspired activation function as a nonlinear mapping function, NMDA$(x):=\phi(x)$. By substituting Eq. 15 into Eq. 18, we show the generalized expression of NMDA$(x)$ equation with $\alpha$ and $\beta$ parameters as following

$$\begin{aligned}
\mathrm{NMDA}_{\alpha,\beta}(x) &= x\mathbf{p}_{\alpha,\beta}(x) \\
&= \frac{x}{1 + \alpha e^{-\beta x}}.
\end{aligned} \tag{19}$$

Given $\alpha = 1$ and $\beta = 1$, $\mathbf{p}(x)$ is identical to sigmoid function, $\sigma(x) = 1/(1 + e^{-x})$. This particular case of $\alpha$ and $\beta$ leads to $x\sigma(x)$, Sigmoid Linear Unit (SiLU) activation function Elfwing et al. (2018). In the case of $\alpha = 1$ and $\beta = 1.702$, $x\sigma(1.702x)$ correspond to the GELU activation function, which is popular in transformer models Lan et al. (2019); Liu et al. (2019); Lan et al. (2019). Ramachandran et al. (2017) introduced the swish activation function, $x\sigma(\beta x)$, which is a generalized form of GELU and SiLU. They demonstrated that when $\beta \to \infty$, the activation function resembles RELU. We summarized these four activation functions by comparing them with our NMDA$_{\alpha,\beta}(x)$ in table 1 and Fig 6.

In contrast to the extensive research on $\beta$ in NMDA$_{\alpha,\beta}(x)$, $\alpha$, the Mg$^{2+}$-gating component, is not explored. For this reason, we focused on the parameter $\alpha$ over $\beta$, and investigated NMDA$_{\alpha}$(X). It is interesting to note that the Swish function was originally proposed as a self-gating function, inspired by the use of the sigmoid function as a gating of information flow in the long short-term memory (LSTM) network Hochreiter & Schmidhuber (1997). In contrast, our activation function NMDA$(x)$ is inspired by the physical Mg$^{2+}$-gating mechanism that occurs in the real biological synapses. These shared mechanisms of self-gating from the artificial models and biological observations raise an interesting possibility that NMDAR is a neural substrate of nonlinear activation function in the brain.

## A.3  DETAILED DESCRIPTION OF TASK DESIGN AND DEFINITION OF SHORT-TERM WORKING MEMORY AND LONG-TERM REFERENCE MEMORY

Our task is based on a widely employed neuroscience experiment for spatial working memory and reference memory Olton et al. (1977; 1979). Errors in working memory are measured by within-trial error, whereas errors in reference memory are measured by across-trial error. The training phase and

Table 1: Comparison of common activation functions with NMDA$_{\alpha,\beta}$

| NMDA$_{\alpha,\beta}$ | Name | Equation | Reference |
|---|---|---|---|
| NMDA$_{\alpha=1,\beta=1}(x)$ | SiLU(x) | $x\sigma(x)$ | Elfwing et al. (2018) |
| NMDA$_{\alpha=1,\beta=1.702}(x)$ | GELU(x) | $x\sigma(1.702x)$ | Hendrycks & Gimpel (2016) |
| NMDA$_{\alpha=1,\beta=\infty}(x)$ | RELU(x) | $\max(0, x)$ | Nair & Hinton (2010) |
| NMDA$_{\alpha=1,\beta}(x)$ | Swish(x) | $x\sigma(\beta x)$ | Ramachandran et al. (2017) |

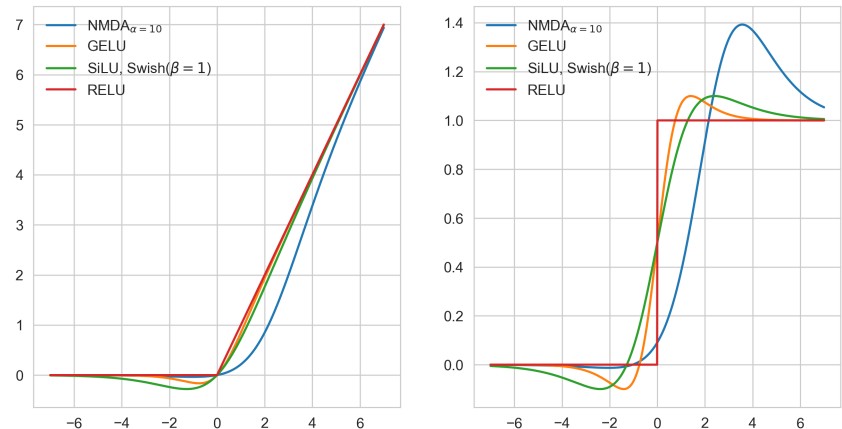

Figure 6: Comparison of common activation functions (left) and their derivatives (right) with $\mathrm{NMDA}_{\alpha,\beta}$.

the test phase alternate at each trial. In the test phase, the unvisited place prediction error and visited place prediction error for the familiar map and the novel map, respectively, are measured. The memory of a relatively recent experience can be defined as *short-term working memory* (STWM), and the memory of relatively old experience can be defined as *long-term reference memory* (LTRM). Within trial visited place prediction measures relatively short-term experience for our task. On the other hand, the across-trial unvisited place prediction task in the familiar map measures the relatively long-term experience. Measuring unvisited place prediction error in the novel map will establish a baseline of chance-level accuracy; above this baseline, the formation of long-term memory can be observed (Fig. 7).

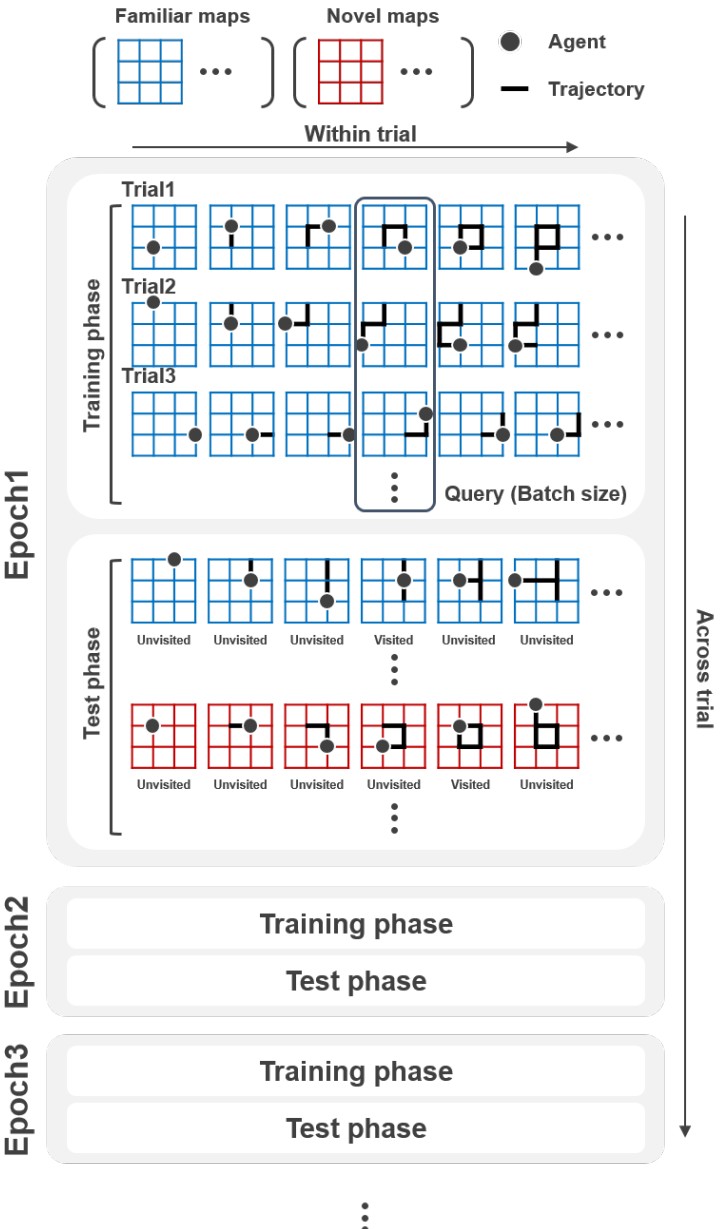

Figure 7: Detailed task design of working and reference memory evaluation. At each random walk step, a batch is created (which is then used in the backpropagation step). The batch size is 512 since there are 512 parallel random walkers in use. Note that at each trial the agent randomly selects a map from training maps (familiar maps), the initial position of the agent is random, and the agent does a random walk.

### A.4 NON-RECURRENT POSITIONAL EMBEDDINGS AND PREDICTION ERRORS ON THE NODE VISITED FOR THE FIRST TIME

We test the non-recurrent positional embedding by substituting the recurrent positional embedding $\mathbf{e}_t$ with the action embedding $A(a_t)$, where $A$ is the embedding layer and $a_t$ is the action at step $t$. Compared to Fig. 8a, the result sdemonstrates a significant increase in working memory error and reference memory error (Fig. 3 vs. Fig. 8). Nonetheless, its behavior is comparable to the trend of decreasing reference memory error while increasing $\alpha$ of NMDA$_\alpha$ (see Fig. 8b).

In addition, we compare unvisited node prediction error (unvisited within context window, in this case, 64 steps) versus first visited node prediction error (unvisited for within a trial). As shown in Fig. 9, the prediction error results for the first visited node do not differ from the reference memory error results.

These results strongly support that (1) while the path-integrated information from recurrent positional embedding is important for learning the spatial structure of the map, this information is not used in predicting the unvisited node, and (2) the reference memory is used for predicting the unvisited node in a familiar map.

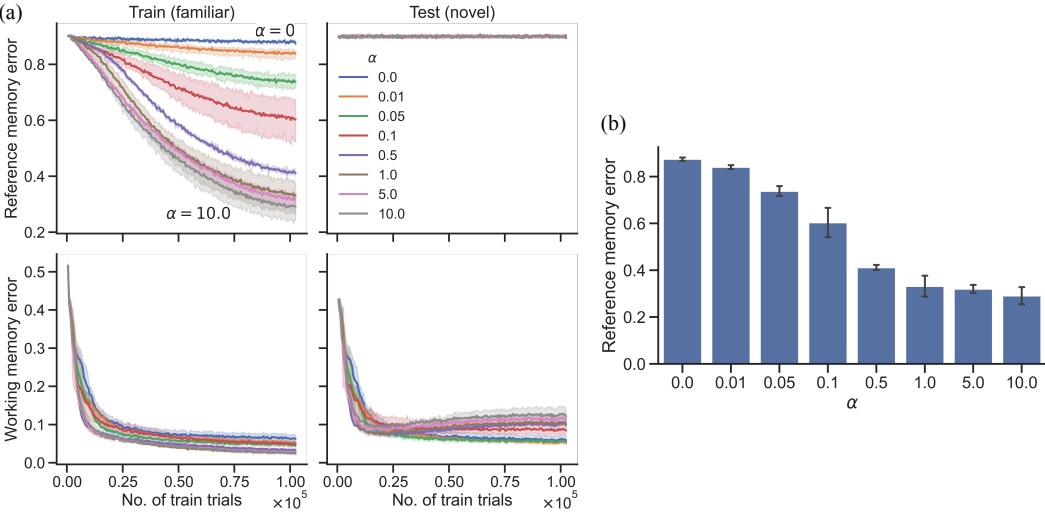

Figure 8: Experiment with non-recurrent positional embeddings. (a) Reference and working memory errors over training trials for training (familiar) maps and testing (novel) maps for $N = 32$ where $N$ is the number of training maps. (b) Reference memory errors evaluated on training maps over different values of $\alpha$ in NMDA$_\alpha$ for $N = 32$. Error bars and shaded areas represent the standard deviation of errors from three independently trained models.

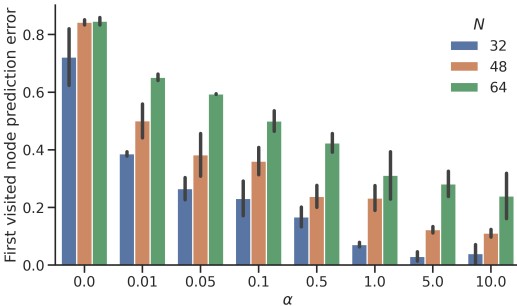

Figure 9: First visited node prediction error evaluated on training maps over different values of $\alpha$ in NMDA$_\alpha$ for $N = 32, 48$, and $64$. Error bars and shaded areas represent the standard deviation of errors from three independently trained models.

## A.5 Transformer as a Memory Consolidation Model and its Biological Plausibility

Next, we examined the biologically inspired $NMDA_\alpha$ activation function in the feed-forward layer of the transformer and its role in memory formation and place cell representation. We show that modulating $\alpha$ corresponds to a change in extracellular $[Mg^{2+}]$, by deriving the nonlinear activation function from the real NMDAR nonlinear IV curve. We show the reconstructed real nonlinear IV curve in Fig. 1a (right top).

The modulation of $\alpha$ selectively affects the formation of long-term reference memory (i.e., prediction of unvisited places across trials) while leaving the formation of short-term working memory (i.e., prediction of unvisited places within trials) intact. This result suggests that short-term working memory and long-term reference memory are physically stored in separate structures: the self-attention layer and the feed-forward layer. In psychology, the idea of a multi-store model regarding short-term memory and long-term memory was historically suggested in Atkinson & Shiffrin (1968). In their model, sensory inputs are stored in short-term memory systems via attention, and some are transferred to a long-term memory system while others quickly disappear.

In neuroscience, the transfer of short-term memory into a long-term system is called *memory consolidation* McGaugh (2000). Animal studies have demonstrated that the CA1 region of the hippocampus is essential for memory consolidation Shimizu et al. (2000); Remondes & Schuman (2004). In hippocampal CA1, the postsynaptic NMDA receptor mediates synaptic plasticity, and the selective perturbation of these receptors leads to impairment in long-term memory formation Tsien et al. (1996); Remondes & Schuman (2004). Later research revealed that $Mg^{2+}$-gating of NMDA receptors modulates the formation of long-term memor Slutsky et al. (2010); Miyashita et al. (2012). These observations imply that the nonlinear dynamics of NMDA receptors in CA1 are critical for consolidating short-term memory into long-term memory.

**On the basis of a previous link between the hippocampus and the transformer, the current research hypothesized that the transformer could serve as a model for memory consolidation.** Given the resemblance of the GELU nonlinear activation function and CA1 NMDAR nonlinear IV curve, we assumed that the GELU activation function serves as a key component that links short-term working memory and long-term reference memory. Our experimental results indicate that that the formation of long-term reference memory is impaired when the activation function is completely linear (corresponding to no $Mg^{2+}$). In contrast, increasing $\alpha$ (which corresponds to an increase in $Mg^{2+}$ level) has resulted in our model's superior performance in long-term reference memory compared to other activation functions (e.g., RELU, GELU, LRELU, Sigmoid, Tanh). Based on these similarities between hippocampal memory consolidation and our results, we propose a transformer as an effective memory consolidation model.

In addition to the performance gain in long-term memory formation with $NMDA_\alpha$, we found that modulating the $\alpha$ affects the emergences of place cells in the feed-forward layer and conclude a significant correlation between place cell score and long-term reference memory formation. Our results align with previous biological findings that perturbation of CA1 NMDARs lead to impairment in both place cell representation and long-term memory formation Tsien et al. (1996); McHugh et al. (1996); Kentros et al. (1998); Shimizu et al. (2000). These similarities support the idea that place cells are the neural correlates of long-term spatial memories. Altogether, our findings suggest an exciting possibility that the nonlinear IV curve of NMDAR in the hippocampal CA1 is a neural substrate of nonlinear activation function in the brain.

## A.6 PSEUDO CODE FOR CALCULATING PLACE CELL SCORE METRIC

---

**Algorithm 1:** Pseudo code for calculating place cell score metric

---

1  function PlaceCellScore(place field);
   **Input** : place field ($K \times K$ 2D array)
   **Output:** place score
2  $G :=$ 2D grid graph ($K \times K$)
3  $\mathcal{G} :=$ empty directed graph ($K \times K$)
4  **for** *edge ($node_i \rightarrow node_j$) in* **G do**
5      **if** *firing rate $\rho_i >$ firing rate $\rho_j$* **then**
6         |   $\mathcal{G}$ add *edge ($node_i \rightarrow node_j$)*
7      **end**
8  **end**
9  Find $node_k$ of firing rate $\rho_{max}$
10 **for** $node_v$ *in* $\mathcal{G}$ **do**
11     **if** $node_v$ *is not descendant of $node_k$ found with DFS(k)* **then**
12        |   delete $node_v$ from $\mathcal{G}$
13     **end**
14 **end**
15 conn. components = sum of all nodes' firing rates in $\mathcal{G}$
16 total components = sum of all nodes' firing rates in $G$
17 **return** place score place score $= \gamma \dfrac{\text{conn. components}}{\text{total components}}$
18 $^{\dagger}$ $\gamma$ is discount factor, determined by connected component size

---

The place field in Algorithm 1 is measured as following procedure: 1) During a random walk simulation, the activation value of a neuron at node $i$, where the agent is located, is measured every 65 steps. Let's say this value is $a_i$. 2) Every time the agent visits node $i$ again, value $a_i$ is added cumulatively to the recorded value; $A_i + = a_i$ such that $A_i$ is the cumulative activation value at node $i$. We assume the initial value of $A_i$ is zero. After the random walk is done, $A_i$ divided by the length of the random walk trajectory is the firing rate $\rho_i$ at node i of the neuron (place field $\in \mathbb{R}^{K \times K}$). In our place cell evaluation experiment, the length of the random walk is $10^5$ and $K = 11$; the evaluate map is one of the training maps.

## A.7 CONSEQUENT OF CHANGING NONLINEAR DYNAMICS IN LEAKY RELU ACTIVATION FUNCTION

Here, we investigated the consequence of changing nonlinearity with other than $\text{NMDA}_\alpha$. Here, we choose LeakyReLU ($\max(0, x) + \alpha \min(0, x)$) activation function to compare with $\text{NMDA}_\alpha$. Regarding LeakyReLU, $\alpha = 1$ of LeakyReLU also leads to a fully linear activation function similar to $\alpha = 0$ of $\text{NMDA}_\alpha$. Compared to $\text{NMDA}_{\alpha=10}$, LeakyReLU exhibits a lower average place score in the allowed range of $\alpha$, indicating that $\text{NMDA}_\alpha$ is better in place cell emergence (see 10)

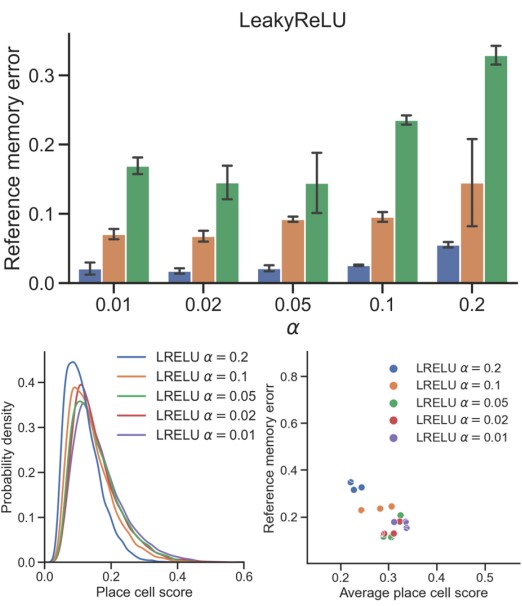

Figure 10: Evaluation of reference memory error in LeakyReLU (LRELU) while modulating $\alpha$(top) and relationship of average place cell score and reference memory error (bottom).

