# OpenReview forum: "Transformer needs NMDA receptor nonlinearity for long-term memory"
_ICLR.cc/2023/Conference — Submitted to ICLR 2023_

### Official Review · Reviewer_o25e · 2022-10-24

**Confidence:** 3
**Correctness:** 2
**Technical Novelty And Significance:** 4
**Empirical Novelty And Significance:** 3
**Recommendation:** 6

**Clarity, Quality, Novelty And Reproducibility:**

### Clarity

Several points seems unclear:

- In the 2.2 and 3.1 subsections, the task is insufficiently detailed. How are the trials conditioned in batches (a figure would be highly appreciated)? What loss function is being used? How are the validation and test sets created and used?
- Page 3: “the initial positional embedding $e_1$ is sampled from a normal distribution”. Can you explain this choice? Why not using a special token?
- Page 4, subsection 3.1, you call $e$ a “positional embedding” whereas you introduced it as an action embedding.
- Page 4: “We run 512 random walk simulations in parallel for collecting training trajectories. The total number of random walking steps is 2,048 for each simulation so the total number of gradient steps for each run is 512 (batch size) × 2,048 (total number of steps in a trial) × 200 (number of trials)”. This part is quite unclear but a better description of the task (as already suggested) would solve it.
- Page 5, subsection 3.3: The explanations are given in reverse:
    - you explain the environment on which you measure the place cell score before defining the place cell score;
    - you define the place cell score before introducing the variables;
    - the notion of firing rate $\rho_i$ is undefined in the paper.
- Page 5, subsection 3.3: “by defining a $K \times K$ 2D grid environment”. It seems implicit but it is new, isn’t it?

### Quality

The figures are explanatory and more of them would be highly appreciated. Further work on the topic could give valuable understanding on the comparison between the brain and artificial neural networks.

### Novelty

The comparison between the GELU activation function and NMDAR dynamics is novel to the best of my knowledge and researches and this model could benefit to the computational neuroscience community.

That being said, other work in computational neuroscience might have tackled this modelling and did not appear when I researched for them. This paper would highly benefit from the review of an expert in neuroscience.

### Reproducibility

The code runs, which is already good sign of reproducibility. More time would be required to determine complete reproducibility. The code is well written and answers a few questions.

**Strength And Weaknesses:**

### Strengths

- The link between GELU and NMDAR dynamics modulated by the concentration $[Mg^{2+}]$ is highly interesting.
- Their model relies on state-of-the-art computational modelling of neurological functions

### Weaknesses

- Several parts of the paper are unclear (details in the next field).
- The main conclusion of the experiments and article is too extrapolated:
“Our data indicated that **NMDAR-like nonlinearity** in the feed-forward network layer of transformers **is necessary** for long-term memory and spatial place cell representation.”
The paper explores what they defined as the $NMDA_{\alpha}$-family of nonlinear function only. It appears that the conclusion drawn is based on the comparison when $\alpha=0$ vs. $\alpha > 0$. Since the former case corresponds to a linear activation function, the conclusion to be drawn from the experiments is that such activation function prevents from place cells-like structures and long-term memory to appear. To sustain their claim, the authors should have compared different nonlinear activation functions with their $NMDA_{\alpha}$. As such, the main claim of the article does not seem sufficiently supported.
- The choice to consider “a node that the agent had never visited within recent 64 steps is treated as an unvisited node.” (page 4) seems lacking. A node visited during a trial on which the model has done a gradient step can be considered as depending on the reference memory, but what if the node is visited for the first time? The model would predict at random level, so no memory could be considered involved.
- Page 4, on the Figure 3: “This finding suggests that the reference memory is non-active for predicting the visited nodes on novel maps.” I find this claim to be confusing. If the test maps are novel and **really** used as a test set and no optimisation step is performed, there is no way the model can integrate information on that map, hence no way for the model to develop a reference memory of the test map. The model can only perform at chance level when confronted to a new environment.
- Page 6, section 4: “On a related note, Whittington et al. (2022) showed that softmax neurons in the self-attention layer behave like place cells and demonstrated that changing the softmax function to linear slows the learning process in the working memory.” But your experiment shows no particular appearance of place cells-like structure in the self-attention layers, right? On what condition do they appear in the feed forward layers rather than in self-attention layers? Without clarification, it feels like place cells can appear anywhere.

More of a suggestion rather than a weakness:

- Giving the number of reference memory error vs. working memory error in addition to the rates (as given in Figure 3) would be informative.

**Summary Of The Paper:**

This paper proposes to model the influence of the concentration of magnesium ions on the $Mg^{2+}$-gated NMDA (a neurotransmitter) receptors nonlinear dynamics - involved in several functions, especially place cells representations,  which are considered important for spatial navigation. The NMDAR nonlinearity is modeled with a GELU-like function. This paper experimentally shows on a 2D grid exploration task how this nonlinearity allows for the appearance of place cells-like structures in the feed forward network of a transformer model.

**Summary Of The Review:**

While this work present highly interesting ideas for the computational neuroscience community, several points are unclear which hinders the understanding of the experiments. Also, the main claim is not sufficiently supported by the experiments. I would be more than glad to increase the score for this paper once the clarifications are made and the main claim is firmly supported (running the experiments with ReLU/tanh/sigmoid/leaky ReLU). Or defer my judgement to an additional reviewer from the neuroscience field.

EDIT:
After clarification, and changing the claim, the authors correctly addressed my concerns and correctly support their work. I hence increased the score.

---

> ### Author Response · Authors · 2022-11-13
> **Author response to reviewer o25e [4/4]**
>
> **4.10 comment**
> >*Page 5, subsection 3.3: The explanations are given in reverse:
> you explain the environment on which you measure the place cell score before defining the place cell score;
> you define the place cell score before introducing the variables;
> the notion of firing rate  is undefined in the paper.*
>
> As reviewer pointed out, our description of place cell score is described in mixed order without clearly introducing variables first. To resolve this description we revised the manuscript as following.
>
> (before)
> We investigate the role of neurons in the FFNs and self-attention layers by measuring the neuron's place specificity. We measure the place cell score by defining a $K \times K$ 2D grid environment as graph $G=(V, E)$ and building a sub-graph $\mathcal{G}=(\mathcal{V}, \mathcal{E})$ of all connected components from the source node $i_{\text{max}}$ where the neuron fires maximally; directed edges of sub-graph $\mathcal{G}$ are generated by connecting high to low firing nodes. We run depth-first-search from $i_{\text{max}}$.
> Given $G$ and $\mathcal{G}$, the place cell score is
> \begin{equation}
> \text{Place cell score} = \gamma\dfrac{ \sum_{i \in \mathcal{V}} \rho_i }{ \sum_{i \in V} \rho_{i} },
> \end{equation}
> where $\gamma=1-|\mathcal{V^*}|/|V|$ is a discount factor and $\mathcal{V^*}$ is a set of nodes from sub-graph without $i_{\text{max}}$ and leaf nodes during depth-first search. $\rho_i$ denotes a firing rate at node $i$. We record the firing rate $\rho_i$ of neurons over a random walking trajectory with $10^5$ steps in one of the training maps; then we measure the place cell scores of neurons in the FFN and self-attention layers. The place cell score is 1 when the neuron is firing only at a certain node; the score is 0 when the neuron is firing homogeneously across all nodes.
>
> (after)
> We investigate the role of neurons in the FFNs and self-attention layers by measuring the neuron's place specificity. Given a $K \times K$ 2D grid environment as graph $G=(V, E)$ and a firing rate (cumulative activation value at node $i$ divided by the length of evaluation trial) of node $i\in{V}$ as a $\rho_i$, we define maximally firing node as $i_{\text{max}}$ and its firing rate as $\rho_{\text{max}}$. Where $E$ is directed edges, which connects high to low firing nodes in $G$. From $G$, we run depth-first-search from source node, $i_{\text{max}}$, to build a sub-graph $\mathcal{G}=(\mathcal{V}, \mathcal{E})$ which we call all connected components.
> Given $G$ and $\mathcal{G}$, the place cell score is defined as following
> $\text{Place cell score} = \gamma\dfrac{ \sum_{i \in \mathcal{V}} \rho_i }{ \sum_{i \in V} \rho_{i} },$
> where $\gamma=1-|\mathcal{V^*}|/|V|$ is a discount factor and $\mathcal{V^*}$ is $\mathcal{V}$ without node $i_{\text{max}}$ and leaf nodes.
> To measure place cell score, we record the firing rate $\rho_i$ of neurons over a random walking trajectory with $10^5$ steps in one of the training maps; then we measure the place cell scores of neurons in the FFN and self-attention layers. The place cell score is 1 when the neuron is firing only at a certain node; the score is 0 when the neuron is firing homogeneously across all nodes.
>
> **4.11 question**
> >*Page 5, subsection 3.3: “by defining a K×K 2D grid environment”. It seems implicit but it is new, isn’t it?*
>
> If we understood reviewer’s comment correctly, reviewer is pointing out that we are defining a new K x K 2d grid environment, which will not have any firing rates if we define a new environment. To resolve this expression, we changed expression as “Given a $K \times K$ 2D grid environment as graph $G=(V, E)$  …"
>
> **References**
> [1] Olton, David S., Christine Collison, and Mary Ann Werz. "Spatial memory and radial arm maze performance of rats." Learning and motivation 8.3 (1977): 289-314.
> [2] Olton, David S., James T. Becker, and Gail E. Handelmann. "Hippocampus, space, and memory." Behavioral and Brain sciences 2.3 (1979): 313-322.
> [3] Banino, Andrea, et al. "Vector-based navigation using grid-like representations in artificial agents." Nature 557.7705 (2018): 429-433.
> [4] Whittington, James CR, et al. "The Tolman-Eichenbaum machine: unifying space and relational memory through generalization in the hippocampal formation." Cell 183.5 (2020): 1249-1263.

---

> > ### Comment · Reviewer_o25e · 2022-11-14
> > **Response to comments 4.10 & 4.11**
> >
> > __4.10 comment__
> >
> > Thank you for changing the order of your explanations. After reading them again, the way the firing rate is defined still puzzle me. If you define them based on a random trajectory with $10^5$ steps, not only will you not explore all the nodes as many times (which will result in differences in the "cumulative activation value at node i", if I understand that term correctly), but also the activation at a given node i should also depend on the context. Wouldn't it have been more fair to sample 65 steps (your whole context length) long sequences ending at node i and compute the firing rate as the activation of the model on each of the sequences? Or did I miss something in the way you defined your firing rate?
> >
> > __4.11 comment__
> >
> > I was more concerned about which memory will your place cells-like structures will rely on. But since Figure 4.e & 4.f report reference memory errors, I guess it must be a map on which you performed training steps, right?

---

> > > ### Author Response · Authors · 2022-11-14
> > > **2nd response to reviewer o25e**
> > >
> > > **4.6 comment**
> > > > *In Figure 7 (which is great, by the way), could you please put in evidence how the batch are formed (a dashed box around a column in the training phase if I understood correctly?)*
> > >
> > > Your observation is correct. At each random walk step, a batch is created (which is then used in the backpropagation step). The batch size is 512 since there are 512 parallel random walkers in use. For better readability, we have included a dashed box on a column in the training phase in Fig. 7.
> > >
> > > > *In the 4.1 subsection: "We run 512 random walk simulations in parallel for collecting training trajectories." At my first reading I had a hard time understanding where did the simulations came from. Could you please add the mention that each simulation is ran on one of the $N$ maps (if it is not already mentioned, in which case I am sorry for the irrelevant remark).*
> > >
> > > We ran each simulation on a randomly selected map among the $N$ maps. As a result, 512 maps are randomly chosen from the N training maps for random walk simulations. Thank you for this question. We found this information to be missing in the manuscript, and now we have revised Section 3.2 as follows:
> > > “An agent starts at a random position and initiates a random walk on the map, __a randomly selected map among $N$ training maps__, for 2,048 steps for each trial.”
> > >
> > > **4.10 comment**
> > > > *Thank you for changing the order of your explanations. After reading them again, the way the firing rate is defined still puzzle me. If you define them based on a random trajectory with 10^5 steps, not only will you not explore all the nodes as many times (which will result in differences in the "cumulative activation value at node i", if I understand that term correctly), but also the activation at a given node i should also depend on the context. Wouldn't it have been more fair to sample 65 steps (your whole context length) long sequences ending at node i and compute the firing rate as the activation of the model on each of the sequences? Or did I miss something in the way you defined your firing rate?*
> > >
> > > Thank you for asking this. We have a similar setup; we sampled a 65-step-long sequence in a single long trajectory ($10^5$  steps). We measured the activation value of a neuron at node $i$ on which the agent is, every 65 steps. Let's say this value is $a_i$. Every time an agent visits node
> > > $i$ again, we add this value cumulatively to the recorded value; $A_i += a_i$ such that $A_i$ is the cumulative activation value at node $i$. We assume the initial value of $A_i$ is zero.
> > > We have included the above information in the manuscript (see updated Appendix A.6).
> > >
> > > **4.11 comment**
> > > > *I was more concerned about which memory will your place cells-like structures will rely on. But since Figure 4.e & 4.f report reference memory errors, I guess it must be a map on which you performed training steps, right?*
> > >
> > > Yes, it is correct. The reference memory errors in Fig 4.e and Fig 4.f are the same as in Fig. 3b (i.e., reference memory errors in training maps). We have included this information in the caption for Fig. 4 as “All results are evaluated from training maps.”
> > >
> > > We appreciate your quick response and we will be pleased to address any additional inquiries.

---

> ### Author Response · Authors · 2022-11-13
> **Author response to reviewer o25e [3/4]**
>
> **4.4 comment**
> >*Page 6, section 4: “On a related note, Whittington et al. (2022) showed that softmax neurons in the self-attention layer behave like place cells and demonstrated that changing the softmax function to linear slows the learning process in the working memory.” But your experiment shows no particular appearance of place cells-like structure in the self-attention layers, right? On what condition do they appear in the feed forward layers rather than in self-attention layers? Without clarification, it feels like place cells can appear anywhere.*
>
> Thank you for this insightful question. Our work showed that place cells emerge both in the self-attention layers and feed-forward networks (see Fig. 4). We reported that the nonlinear $\alpha$ value in feed-forward networks does not affect the emergence of place cells in the self-attention layers. In our picture, we think both the self-attention layer and the feed-forward layer have spatial representations of place cells, while they represent short-term working memory and long-term reference memory respectively.
>
> **4.5 comment**
> >*Giving the number of reference memory error vs. working memory error in addition to the rates (as given in Figure 3) would be informative.*
>
> This is a great suggestion. The average number of unvisited nodes in a single trial is 561. Our updated manuscript now includes this information.
>
> **4.6 question**
> >*In the 2.2 and 3.1 subsections, the task is insufficiently detailed. How are the trials conditioned in batches (a figure would be highly appreciated)? What loss function is being used? How are the validation and test sets created and used?*
>
> We used softmax cross-entropy loss for training and predicted the subsequent sensory observations (i.e., dashed squares), as described in the Method section. We have substantially revised the experiment's description to add details. We hope our update answers your question.
>
> **4.7 question**
> >*Page 3: “the initial positional embedding $e_1$ is sampled from a normal distribution”. Can you explain this choice? Why not using a special token?*
>
> $e_1$ is the initial hidden state of the RNN. If the initial location of the agent is given or fixed, then using a special token makes more sense. However, the agent is initially randomly placed and no initial position information is given in our task, so we sampled $e_1$ from a normal distribution as a prior on initial location.
> In related works, Banino et al. [3] used the LSTM model for position prediction tasks and they employed special tokens for each initial location; they provided the special token of the initial position as an initial hidden state of LSTM. On the other hand, Whittington et al. [4] did not provide initial location information to the model, and $e_1$ is randomly sampled from a prior.
>
> **4.8 comment**
> >*Page 4, subsection 3.1, you call $e_t$ a “positional embedding” whereas you introduced it as an action embedding.*
>
> Thank you for pointing this out. We would like to clarify that we introduced $e_t$ as “positional embedding” in the Method section but not as "action embedding." Perhaps to clarify the meaning better, we have changed all mentions of $e_t$ to “recurrent positional embedding”. We hope this change clarifies the description.
>
> **4.9 comment**
> >*Page 4: “We run 512 random walk simulations in parallel for collecting training trajectories. The total number of random walking steps is 2,048 for each simulation so the total number of gradient steps for each run is 512 (batch size) × 2,048 (total number of steps in a trial) × 200 (number of trials)”. This part is quite unclear but a better description of the task (as already suggested) would solve it.*
>
> To clarify the obscure expression, we added a detailed description of the task in Appendix A.3 with new Fig. 7.

---

> > ### Comment · Reviewer_o25e · 2022-11-13
> > **Comments 4.[4-9]**
> >
> > __4.4 comment__
> >
> > Thank you for your response, I might have misunderstood something in the first version and it now appears much clearly.
> >
> > __4.5 comment__
> >
> > Thank you for adding this, it seems interesting for a fair comparison in reference errors.
> >
> > __4.6 comment__
> >
> > The added description and figures help. A few more details (sorry if that seems annoying):
> > - In Figure 7 (which is great, by the way), could you please put in evidence how the batch are formed (a dashed box around a column in the training phase if I understood correctly?)
> > - In the 4.1 subsection:
> > > We run 512 random walk simulations in parallel for collecting training trajectories.
> > At my first reading I had a hard time understanding where did the simulations came from. Could you please add the mention that each simulation is ran on one of the N maps (if it is not already mentioned, in which case I am sorry for the irrelevant remark).
> >
> > __4.7 comment__
> >
> > Thank you for giving explanations regarding this choice.
> >
> > __4.8 comment__
> >
> > The change for "recurrent positional embedding" seems fine and clear, thank you.
> >
> > __4.9 comment__
> >
> > This concern has been addressed in comment 4.6. Thanks again.

---

> ### Author Response · Authors · 2022-11-13
> **Author response to reviewer o25e [2/4]**
>
> **4.2 comment**
>
> > *The choice to consider "a node that the agent had never visited within recent 64 steps is treated as an unvisited node." (page 4) seems lacking. A node visited during a trial on which the model has done a gradient step can be considered as depending on the reference memory, but what if the node is visited for the first time? The model would predict at random level, so no memory could be considered involved.*
>
> We measure the reference memory error based on the prediction error on node that the agent had not visited in the past 64 steps; in this case, the model could use the information in recurrent position embeddings for the prediction instead of using reference memory. So we compared the unvisited place pediction error (unvisited within context window, here 64 step) vs. first visited place prediction error (i.e. unvisited for within a trial; the case you suggested). We evaluate the first visited node prediction error, and the results show no difference from the reference memory error results in training maps. This result support that reference memory is involved for the unvisited node prediction. We have included this finding in the Appendix A.4 (see Fig. 9).
>
> **4.3 comment**
> > *Page 4, on the Figure 3: “This finding suggests that the reference memory is non-active for predicting the visited nodes on novel maps.” I find this claim to be confusing. If the test maps are novel and **really** used as a test set and no optimisation step is performed, there is no way the model can integrate information on that map, hence no way for the model to develop a reference memory of the test map. The model can only perform at chance level when confronted to a new environment.*
>
> Thank you for reviewer’s thoughtful comment. We are sorry to make confusing claim to reviewer, “This finding suggests that the reference memory is non-active for predicting the visited nodes on novel maps.”. As the other reviewer also pointed out, we find this claim is not valid, as reference memory cannot be defined in the novel map at first place. To clearly state the definition of unvisited place prediction, we added following contents in Appendix A.3:
>
> "Appendix A.3: Detailed task design of working and reference memory evaluation
> Our task is based on a widely employed neuroscience experiment for spatial working memory and reference memory [1, 2]. Errors in working memory are measured by within-trial error, whereas errors in reference memory are measured by across-trial error. The training phase and the test phase alternate at each trial. In the test phase, the unvisited place prediction error and visited place prediction error for the familiar map and the novel map, respectively, are measured. The memory of a relatively recent experience can be defined as *short-term working memory* (STWM), and the memory of relatively old experience can be defined as *long-term reference memory* (LTRM). Within trial visited place prediction measures relatively short-term experience for our task. On the other hand, across trial unvisited place prediction task in the familiar map measures the relatively long-term experience. Measuring unvisited place prediction error in the novel map will establish a baseline of chance-level accuracy; above this baseline, the formation of long-term memory can be observed (see Fig. 7 in our revised manuscript)."
>
> We also revised the text as “working memory formation is intact on novel maps”, as that is the best description we can find in the test (novel) map result. I hope this explanation clears up the confusion.

---

> > ### Comment · Reviewer_o25e · 2022-11-13
> > **Response to comment 4.2 & 4.3**
> >
> > __4.2 comment__
> >
> > Thank you for your thorough response.
> > > So we compared the unvisited place prediction error (unvisited within context window, here 64 step) vs. first visited place prediction error (i.e. unvisited for within a trial; the case you suggested).
> > This is not what I had in mind when I wrote the review but the way you defined "first visited place" is exactly what I would have thought reference memory should have been evaluated when reading at its description. Your complementary experiment seems highly relevant to me.
> >
> > __4.3 comment__
> >
> > The changes here address my concerns. Thank you for your explanations.

---

> ### Author Response · Authors · 2022-11-13
> **Author response to reviewer o25e [1/4]**
>
> **Overall Response**
>
> The authors thank the reviewers for insightful feedback that helped improve our work. Please find our response to each comment below. We have also made substantial changes to the manuscript. We will be happy to follow up on additional feedback.
>
> **4.1 comment**
>
> > *The main conclusion of the experiments and article is too extrapolated: “Our data indicated that NMDAR-like nonlinearity in the feed-forward network layer of transformers is necessary for long-term memory and spatial place cell representation.” The paper explores what they defined as the NMDAα-family of nonlinear function only. ... To sustain their claim, the authors should have compared different nonlinear activation functions with their NMDAα. As such, the main claim of the article does not seem sufficiently supported.
> Comment from summary of the review: ... I would be more than glad to increase the score for this paper once the clarifications are made and the main claim is firmly supported (running the experiments with ReLU/tanh/sigmoid/leaky ReLU).*
>
> We sincerely thank the reviewer for the encouragement and the suggestion to test with nonlinear activation functions. Figure 3c in the revised manuscript displays the new results. As shown in the additional results, our NMDA activation function with $\alpha=10$ has the lowest reference memory error compared to other nonlinear functions.
>
> Based on these new results, we have toned down the statements, for example, to
>
> “Our data indicated that NMDAR-like nonlinearity in the feed-forward network layer of transformers can enhance the formation of long-term memory and spatial place cell representation. Furthermore, this design choice improves long-term memory more than other commonly used nonlinear functions.”

---

> > ### Comment · Reviewer_o25e · 2022-11-13
> > **Response to comment 4.1**
> >
> > Thank you for your helpful reply.
> > The now toned down statement seems adapted and well supported by your experiments.

---

> ### Author Response · Authors · 2022-11-27
> **Dear reviewer o25e**
>
> We thank you for your thoughtful reviews and your responses. We have improved our manuscript based on your comments and it would be great if you give us further feedback on our revised version of the manuscript.

---

### Official Review · Reviewer_7x8r · 2022-10-24

**Confidence:** 3
**Correctness:** 1
**Technical Novelty And Significance:** 2
**Empirical Novelty And Significance:** 3
**Recommendation:** 3

**Clarity, Quality, Novelty And Reproducibility:**

**Clarity**

Several aspects of the paper are unclear:
* What do the NMDAR IV curves in Figure 1 a) correspond to? The NMDAR open probability lacks a factor of $x$ if it is supposed to model the behavior in the figure.
* The paper does not justify why the positional encodings, which are particularly relevant in a 2D navigation setup, are omitted.
* The inset working memory plot in Figure 3 a) is quite confusing and should be presented as a standalone figure.
* Figure 3 b) does not specify whether the reference memory error is measured on train or test maps.
* An illustration of the place cell score computation would help the reader understand the metric.

**Quality**

I am not sure if ICLR is the appropriate venue for the paper, given both its topic and length (i.e., 6.5 pages).

**Novelty**

To the best of my knowledge, the proposed activation function and connection to the NMDAR have not been investigated before.

**Reproducibility**

Given the lack of clarity of the experimental setup (see above), I am not convinced that the results would be entirely reproducible.


**Strength And Weaknesses:**

**Strengths**

The paper draws on insight from neuroscience to explain the behavior of the transformer architecture, which is interesting. Moreover, the experimental evaluation showcases an interesting connection between the sparsity of connections in the feed-forward network of the transformer and the proposed activation function.

**Weaknesses**

The two considered memory types are ill-defined, leading to void claims:
* In the case of working memory, any nonlinearity should be capable of decreasing the working memory error on test maps (as evidenced by the fact that the error is largely independent of the proposed activation function’s hyperparameter) due to the global attention mechanism of transformers (i.e., as long as the observation is within the context window). Thus, the connection to NDMARs is tenuous.
* In the case of reference memory, evaluating the model on unseen maps is an ill-defined problem, since the unvisited places are inherently unpredictable. Thus, claiming that `reference memory is non-active for predicting the visited nodes on novel maps` or that `reference memory formation requires NMDAR-like nonlinearity` does not make sense.

The place cell analogy is tenuous, since the proposed score only measures sparsity and not the location in the 2D grid environment, unlike place cells. The results on the sparsity are still interesting, but the claim relating them to place cells has to be revised.


**Summary Of The Paper:**

The paper draws connections between the NMDA receptor (NMDAR) in the hippocampus and the GELU activation function, which has been employed in the transformer architecture. The paper then proposes a novel activation function, which more closely resembles the behavior of the NMDAR, and shows that the transformers memory capabilities can be tuned with the hyperparameter of this activation function. In particular, the paper investigates the working memory (i.e., in-context memory of states observed during the current trajectory) and the reference memory (i.e., out-of-context memory of states observed during previous trajectories) of the transformer on a 2D grid navigation experiment. Finally, the paper also proposes a place cell score, which measures the sparsity of a neuron’s connections, and shows that it can also be tuned with the hyperparameter of the proposed activation function.

**Summary Of The Review:**

Given the ill-defined memory metrics and tenuous connection to place cells, I do not recommend the paper for acceptance at this time.

---

> ### Author Response · Authors · 2022-11-13
> **Author response to reviewer 7x8r [2/2]**
>
> Among them, the Peak method is the closest form of our place cell metric, which is found to be 1) robust to variations in place fields and 2) no inherent assumptions about the spatial information in place fields. As our transformer model learns the relational information of sensory and action pairs but not the precise location information (i.e., x and y coordinates), we wanted to avoid any inherent assumptions regarding the spatial information.
>
> Second, our metric is based on the place cell score metric described in Whittington’s TEM-t model [2], which is also measured in 2D-grid space. As 2D-grid space cannot be directly interpreted as euclidean space, it is not justifiable to apply space-related functions such as gaussian kernel density estimation on the place fields. Thus, we decided to choose the Peak method approach, which can be easily applicable also on the 2D-grid structure.
>
> In summary, we show that our place cell metric is 1) similar to the Peak method metric that is recommended in neuroscience field [1], 2) and based on the previous description [2], which works in the 2D-grid space. Moreover, we provide pseudo-code for place cell metrics to give an intuition for the potential reviewers.
>
> **3.3 question**
> >*What do the NMDAR IV curves in Figure 1 a) correspond to? The NMDAR open probability lacks a factor of x if it is supposed to model the behavior in the figure.*
>
> Figure 1a correspond to Equation 17, $I_{\text{norm}}=\text{V}{\bf p}(\text{V})$ in Appendix A.1 entitled “Derivation of NMDAR nonlinearity from the molecular level chemical interaction”. This equation is reconstructed IV curve from Kirson et al. (1999) [3]. NMDAR IV curve is multiplication of input voltage V and voltage dependent NMDAR open probability ${\bf p}(\text{V})$. To clearly show the NMDAR-nonlinearity inspired activation function derivation, we included two sections in Appendix: Appendix A.1 "Derivation of NMDAR nonlinearity from the molecular level chemical interaction" and Appendix A.2 "NMDAR-inspired nonlinear activation function".
>
> **3.4 comment**
> >*The paper does not justify why the positional encodings, which are particularly relevant in a 2D navigation setup, are omitted.*
>
> Thank you for this suggestion. We did not leave out positional encodings (sinusoidal position encodings in Vaswani et al., (2017) [4]). We replaced these positional encodings with recurrent positional embeddings $e_t$ in the transformer model's input layer. We have modified the sentence in Method section as follows:
> “Instead of using __sinusoidal__ positional encoding [4] that is commonly used in transformers, we employ the __recurrent positional embedding which is encoding the location of an input element by using__ the recurrent neural network (RNN) [2]”
>
> **3.5 comment**
> >*The inset working memory plot in Figure 3 a) is quite confusing and should be presented as a standalone figure. Figure 3 b) does not specify whether the reference memory error is measured on train or test maps.*
>
> Many thanks for your suggestions and for pointing out the missing information. We have updated Fig. 3a by presenting standalone figures for each case. Fig. 3b caption now states that the reference memory error is measured on training maps.
>
> **3.6 comment**
> >*An illustration of the place cell score computation would help the reader understand the metric.*
>
> Thank you for this suggestion to improve the paper. Our newly added pseudo code for place cell score calculation (see Algorithm 1 in the Appendix A.6) which describes in detail how to calculate the place cell score metric. In short, we find the largest connected component from the maximally firing node. The place score is then calculated by dividing the firing mass of the largest connected component by the total firing mass.
>
> **3.7 comment**
> >*Given the lack of clarity of the experimental setup (see above), I am not convinced that the results would be entirely reproducible.*
>
> We included the Pytorch implementation code available along with the paper (see Supplemental Material). We have included new figures in the Appendix describing the experimental setup, task design, and pseudo-code for reproducibility.
>
>
>
>
> **References**
> [1] Grijseels, Dori M., et al. "Choice of method of place cell classification determines the population of cells identified." PLoS computational biology 17.7 (2021): e1008835.
> [2] Whittington, James CR, et al. "Relating transformers to models and neural representations of the hippocampal formation." International Conference on Learning Representations (2022).
> [3] Kirson, Eilon D., et al. "Early postnatal switch in magnesium sensitivity of NMDA receptors in rat CA1 pyramidal cells." The Journal of Physiology 521.Pt 1 (1999): 99.
> [4] Vaswani, Ashish, et al. "Attention is all you need." Advances in neural information processing systems 30 (2017).

---

> ### Author Response · Authors · 2022-11-13
> **Author response to reviewer 7x8r [1/2]**
>
> **Overall Response**
>
> The authors thank the reviewers for insightful feedback that helped improve our work. Please find our response to each comment below. We have also made substantial changes to the manuscript. We will be happy to follow up on additional feedback.
>
> **3.1 comment**
> >*The two considered memory types are ill-defined, leading to void claims:*
> >* *In the case of working memory, any nonlinearity should be capable of decreasing the working memory error on test maps (as evidenced by the fact that the error is largely independent of the proposed activation function’s hyperparameter) due to the global attention mechanism of transformers (i.e., as long as the observation is within the context window). Thus, the connection to NMDARs is tenuous.*
> >* *In the case of reference memory, evaluating the model on unseen maps is an ill-defined problem, since the unvisited places are inherently unpredictable. Thus, claiming that reference memory is non-active for predicting the visited nodes on novel maps or that reference memory formation requires NMDAR-like nonlinearity does not make sense.*
>
> We appreciate the reviewer's frank comment. We made significant changes to the manuscript to clarify our descriptions and to state the task design in detail. In the updated manuscript, all changes are highlighted in blue text. In particular, please see newly added Appendix A.3. We hope that these changes will help fill the gap and any misunderstanding.
>
> We’d also like to address the reviewer’s comment on "The two considered memory types are ill-defined”. First, our definition of working memory is estimated by within trial visited place prediction error, and reference memory is estimated by across trial unvisited place prediction error in “familiar maps”.  We recognize that it could be confusing when we mix the term “reference memory error” on “novel map.” The more accurate description would be “unvisited place prediction error” for both “familiar map” and “novel map”. The measure of “unvisited place prediction error” in the novel map will set the baseline with chance level accuracy, and above this baseline can be regarded as a reference memory.
>
> Thus, we agree that evaluating the model on unseen maps is an "ill-posed" problem, while our main claim is not based on it. Our claim is based on the result of the unvisited place prediction error in the familiar map, which we think of as a valid measure for reference memory assessment. Our main claim that “reference memory formation requires NMDAR-like nonlinearity” comes from the comparison of $\alpha=0$, and $\alpha>0$ from across-trial unvisited place prediction error in familiar maps.
> Regarding the reviewer's comment, the sentence "reference memory is non-active for predicting the visited nodes on novel maps" should be changed because reference memory does not exist if the map is unseen in the first place. We thank the reviewer for careful reading and pointing this out. We have modified the sentence to "working memory formation is intact on novel maps."
>
> **3.2 comment**
> >*The place cell analogy is tenuous, since the proposed score only measures sparsity and not the location in the 2D grid environment, unlike place cells. The results on the sparsity are still interesting, but the claim relating them to place cells has to be revised.*
>
> This feedback is greatly appreciated. We assume the reviewer's concern stems from our metric's lack of location information; thus, our metric appears to measure only sparsity but not place score. We are happy to share our reasoning for our place cell metric.
>
> First, we demonstrate that our metric, the Peak method, is one of the recommended metrics in the neuroscience field, with some advantages. According to previous research on place cell evaluation, there are primarily four types of methods for place cell evaluation [1]:
> 1. Peak method - classifies based on the average rate of firing in one location being higher than in the rest of the environment
> 2. Stability method - classifies with stable firing patterns across locations over time
> 3. Information method - classifies based on the increased amount of spatial information the cells hold about the animal’s location
> 4. Combination method - combined method considering cell’s place fields, including the size, peak and peak of activity

---

> ### Author Response · Authors · 2022-11-27
> **Dear reviewer 7x8r**
>
> We thank you for your thoughtful reviews and your responses. We have improved our manuscript based on your comments and it would be great if you give us further feedback on our revised version of the manuscript.

---

### Official Review · Reviewer_oq3h · 2022-10-25

**Confidence:** 4
**Correctness:** 3
**Technical Novelty And Significance:** 2
**Empirical Novelty And Significance:** 1
**Recommendation:** 3

**Clarity, Quality, Novelty And Reproducibility:**

This paper is very clearly written with original ideas built upon current hippocampal models. Since the code is provided, I assume the work is reproducible.

**Strength And Weaknesses:**

Strengths:
- The paper presents a well-defined problem with clear communication and writing and easily understandable figures. It continues the line of research on the connections between Transformers and models of the hippocampus and takes inspirations from neurobiology to develop a new model for the hippocampus.

Major Weaknesses:
- This paper only used one nonlinearity, which is NMDA-$\alpha$. It would be helpful to include some comparison to other types of nonlinearity commonly used in hippocampal-entorhinal models.
- This paper claims that NMDAR nonlinearity is needed for *long-term memory* when  it only shows the effect of $\alpha$ on reference memory. What about other types of long-term memory, such as episodic memory?
- What are the mathematical implications of replacing GELU with NMDA-alpha? It would strengthen the paper to see a comparison between GELU-based Transformer and NMDA-$\alpha$ based Transformer in terms of behavioral performance and place cell representations.
- Figure 3a shows that the model doesn’t perform well on predicting unvisited node in Novel Map, which, in my understanding, means that the model cannot learn latent structure of the map and do flexible binding, which is what the the original Tolman-Eichenbaum Machine (Whittington et al. 2020) is *supposed* to do. This should be discussed more in details in the discussion section.
- Figure 3b: If I’m understanding correctly, training the model on *more* maps leads to *bigger* errors? Is there an explanation for this?
- There might exist alternative explanations for why the model can predict nodes that haven’t been visited in the past 65 steps in a familiar map, other than reference memory. For example, such behavior could be explained by path integration, which can be attributed to recurrent positional encoding? As such, it might be interesting to see whether non-recurrent positional encoding gives similar results.

Minor weaknesses:
There are some weaknesses inherited from TEM-t:
- The model doesn’t account for some well-established hippocampal phenomenon, eg. replay;
- The sensory prediction task is a toy task with simplified, pre-digested inputs. It’s hard to know whether the model is compatible with other types (eg. image or video) of inputs, or whether it can scale up to work with tasks more suited for current machine learning climate, such as Atari.

Questions:
- TEM-t (Whittington et al. 2022) introduced 3 modifications to the original transformers. If I'm understanding correctly, you didn’t use TEM-t, but Transformer with recurrent positional encoding (that, and NMDA-$\alpha$ instead of GELU). Then how do the other two modifications affect pace cell properties?
- What are the biological implications of changing alpha? The biological parallel of NMDA-$\alpha$ to Mg2+ gated ion channels seemed a little far-fetched to me.


**Summary Of The Paper:**

In this paper, the authors showed that place cells emerge in the feedforward layer of the Transformer that uses 1) NMDA-$\alpha$ nonlinearity and 2) recurrent positional encoding when trained on the sensory observation prediction task. Moreover, they showed that bigger $\alpha$ in the nonlinearity is simultaneously correlated with better reference memory and better place cell scores.


**Summary Of The Review:**

It is a well-written paper with interesting findings and very clear communication, and can benefit the neuro-AI community. However, I believe the scope of this paper is more suited for a workshop or smaller conference. Thus I'd recommend rejecting the paper for ICLR.

---

> ### Author Response · Authors · 2022-11-13
> **Author response to reviewer oq3h [4/4]**
>
> **2.11 summary**
> > It is a well-written paper with interesting findings and very clear communication, and can benefit the neuro-AI community. However, I believe the scope of this paper is more suited for a workshop or smaller conference. Thus I'd recommend rejecting the paper for ICLR.
>
> Thank you for your input. Although the field of neuro-AI is small, we believe it is an important field that is growing fast in both neuroscience and computer science domains. There is a growing body of work published in leading AI venues on deep neural network model analysis and interpretation using a neuroscience approach. Here are a couple of examples: (Cueva et al. ICLR 2018 [10], Sorscher et al. NeurIPS 2019 [11], Whittington et al. ICLR 2022 [6], Chu et al. NeurIPS 2022 [12], Schaeffer et al. NeurIPS 2022 [13]). We hope that our response has persuaded you to change your mind.
>
> **References**
>
> [1] Bird, Chris M., and Neil Burgess. "The hippocampus and memory: insights from spatial processing." Nature Reviews Neuroscience 9.3 (2008): 182-194.
> [2] Henke, Katharina. "A model for memory systems based on processing modes rather than consciousness." Nature Reviews Neuroscience 11.7 (2010): 523-532.
> [3] Olton, David S., Christine Collison, and Mary Ann Werz. "Spatial memory and radial arm maze performance of rats." Learning and motivation 8.3 (1977): 289-314.
> [4] Olton, David S., James T. Becker, and Gail E. Handelmann. "Hippocampus, space, and memory." Behavioral and Brain sciences 2.3 (1979): 313-322.
> [5] Whittington, James CR, et al. "The Tolman-Eichenbaum machine: unifying space and relational memory through generalization in the hippocampal formation." Cell 183.5 (2020): 1249-1263.
> [6] Whittington, James CR, Joseph Warren, and Tim EJ Behrens. "Relating transformers to models and neural representations of the hippocampal formation." International Conference on Learning Representations (2022).
> [7] Devlin, Jacob, et al. "Bert: Pre-training of deep bidirectional transformers for language understanding." arXiv preprint arXiv:1810.04805 (2018).
> [8] Dosovitskiy, Alexey, et al. "An image is worth 16x16 words: Transformers for image recognition at scale." arXiv preprint arXiv:2010.11929 (2020).
> [9] Chen, Lili, et al. "Decision transformer: Reinforcement learning via sequence modeling." Advances in neural information processing systems 34 (2021): 15084-15097.
> [10] Cueva, Christopher J., and Xue-Xin Wei. "Emergence of grid-like representations by training recurrent neural networks to perform spatial localization." International Conference on Learning Representations. 2018.
> [11] Sorscher, Ben, et al. "A unified theory for the origin of grid cells through the lens of pattern formation." Advances in neural information processing systems 32 (2019).
> [12] Chu, Tianhao,  et al. "Oscillatory Tracking of Continuous Attractor Neural Networks Account for Phase Precession and Procession of Hippocampal Place Cells." Advances in neural information processing systems 35 (2022).
> [13] Schaeffer, Rylan, et al. "No Free Lunch from Deep Learning in Neuroscience: A Case Study through Models of the Entorhinal-Hippocampal Circuit." Advances in neural information processing systems 35 (2022).

---

> ### Author Response · Authors · 2022-11-13
> **Author response to reviewer oq3h [3/4]**
>
> **2.7 minor comment**
> > *There are some weaknesses inherited from TEM-t: The model doesn’t account for some well-established hippocampal phenomenon, eg. replay;*
>
> Thank you for sharing this observation. TEM is based on a generative process similar to hippocampal replay. This process is missing in TEM-t, so it cannot explain the hippocampal phenomenon. We are aware of TEM-inherent t's weakness, and our transformer model will not account for replay, as indicated by the reviewer. Designing a replay in our model could be an exciting future work.
>
> **2.8 minor comment**
> > *The sensory prediction task is a toy task with simplified, pre-digested inputs. It’s hard to know whether the model is compatible with other types (eg. image or video) of inputs, or whether it can scale up to work with tasks more suited for current machine learning climate, such as Atari.*
>
> Our sensory prediction task is relatively simple, and it is unclear whether our findings could apply to other tasks. However, since our model is based on the transformer model, our findings can also be tested in the models based on transformers (such as BERT [7], ViT [8], and Decision transformer in RL [9]), by analyzing the performance and the consequences of modulating the NMDA-inspired activation function. Thank you for this feedback. We hope to explore this direction in the future.
>
> **2.9 question**
> > *TEM-t (Whittington et al. 2022) introduced 3 modifications to the original transformers. If I'm understanding correctly, you didn’t use TEM-t, but Transformer with recurrent positional encoding (that, and NMDA-α instead of GELU). Then how do the other two modifications affect pace cell properties?*
>
> For the first modification (NMDA-α instead of GELU), we trained the model with GELU and compared it with NMDA-(α=10). The results show that NMDA-(α=10) has a lower reference memory error and higher place cell scores than GELU.
>
> For the second modification (recurrent positional encoding), we trained the model with non-recurrent learnable position encoding. The result shows that working memory and reference memory errors increase substantially. This finding suggests that recurrent positional encoding is essential for the formation of working memory. It does, however, follow the trend of decreasing reference memory error while increasing $\alpha$ of NMDA.
>
> We have included these results in the updated Fig. 3c and Fig. 4g (GELU), and the impact of the second modification has been added in the Appendix. _Please see A.4, "Non-recurrent positional embeddings and prediction errors on the node visited for the first time"._
>
> **2.10 question**
> > *What are the biological implications of changing alpha? The biological parallel of NMDA-α to Mg2+ gated ion channels seemed a little far-fetched to me.*
>
> We appreciate the reviewer's question and have added a substantial amount of background work describing the biological implications of changing the alpha value. Please see the newly added sections that detail the NMDAR-inspired activation function derived from the molecular level interaction of Mg$^{2+}$ and NMDAR. These sections appear in the Appendix and are titled:
> 1) Section A.1 Derivation of NMDAR nonlinearity from the molecular level chemical interaction
> 2) Section A.2. NMDAR-inspired nonlinear activation function
>
> _Please see our subsections attached in the revised version_

---

> ### Author Response · Authors · 2022-11-13
> **Author response to reviewer oq3h [2/4]**
>
> **2.4 comment**
> > *Figure 3a shows that the model doesn’t perform well on predicting unvisited nodes in Novel Map, which, in my understanding, means that the model cannot learn the latent structure of the map and do flexible binding, which is what the original Tolman-Eichenbaum Machine (Whittington et al. 2020) is supposed to do. This should be discussed more in detail in the discussion section.*
>
> The TEM and TEM-t papers measured the working memory error in a novel map, i.e., prediction error on "visited places" in a novel (test) map. Similarly, our model performs well in terms of working memory (Fig. 3a). The challenging cases are predicting "unvisited nodes" in a novel map, which TEM models also fail at.
>
> Based on the reviewers' comments, we added a new section after the Introduction titled "Transformer" to discuss how our work differs from others (including TEM and TEM-t).
> To summarize, the original Tolman-Eichenbaum Machine (TEM; Whittington et al., Cell 2020) [5] uses the Hebbian matrix as memory storage, which can be changed depending on context (i.e., initial values are zeros in a trial) by using the Hebbian update rule in a single trial. (Whittington et al., ICLR 2022) [6] recently showed that the Transformer’s self-attention mechanism has a similar mathematical formulation to the TEM model and replaced the dynamically changed memory storage, Hebbian matrix, with the self-attention mechanism.
>
> The memory structure is the main distinction between our work and that of TEMs. While the transformer architecture in our work has a fixed length of context window, the context length of TEM-t and TEM models is not fixed, allowing it to keep all previous information in a single trial. This difference explains why our model cannot predict nodes in a novel map that are not in the context (i.e., unvisited nodes in the previous 64 steps).
>
> **2.5 question**
> > *Figure 3b: If I’m understanding correctly, training the model on more maps leads to bigger errors? Is there an explanation for this?*
>
> This is an accurate observation. Training over more maps leads to bigger reference memory errors. This is because more maps require the model to store more pairs of 'what'-'where' memory (i.e., each training contains unique 'what'-'where' information). We added this information in the manuscript.
>
> **2.6 comment**
> > *There might exist alternative explanations for why the model can predict nodes that haven’t been visited in the past 65 steps in a familiar map, other than reference memory. For example, such behavior could be explained by path integration, which can be attributed to recurrent positional encoding? As such, it might be interesting to see whether non-recurrent positional encoding gives similar results.*
>
> Thank you for suggesting an alternative hypothesis. We also had a similar question, which inspired us to test the reference memory (unvisited place prediction error) and working memory errors (visited place prediction error)  on novel (test) maps. If other memory, such as information in path integration (recurrent positional embedding e t), makes predictions on unvisited nodes in a familiar (train) map, then the prediction error on unvisited nodes in novel maps could be minimized because the RNN module can hold positional information. However, Fig. 3a shows that this is not the case.
> We followed up on the proposed idea and trained our model with the non-recurrent learnable positional encoding; the result shows that working memory and reference memory errors increase substantially. However, it exhibits similar behavior to the trend of decreasing reference memory error while increasing $\alpha$ of NMDA.
>
> We also investigated the first visited node prediction error. While we defined the reference memory error as a prediction error on the node that the agent does not visit in the past 65 steps, the first visited node prediction error is a prediction error on the node that the agent had never visited in a single trial. The result on the first visited node prediction error shows no difference from the reference memory error results.
>
> These findings are included in the Appendix. They imply that (1) while path-integrated information from recurrent position embedding is useful for learning the spatial structure of the map, it is not used to predict the unvisited node. (2) In a familiar map, the reference memory is used to predict the unvisited node in the previous 65 steps.

---

> ### Author Response · Authors · 2022-11-13
> **Author response to reviewer oq3h [1/4]**
>
> **Overall Response**
> The authors thank the reviewers for insightful feedback that helped improve our work. Please find our response to each comment below. We have also made substantial changes to the manuscript. We will be happy to follow up on additional feedback.
>
> **2.1 comment**
> >*This paper only used one nonlinearity, which is NMDA-α. It would be helpful to include some comparison to other types of nonlinearity commonly used in hippocampal-entorhinal models.*
>
> Thank you for this feedback. To our knowledge, the most commonly used functions for nonlinearity in hippocampal-entorhinal models are rectified linear unit (ReLU) and LeakyReLU. We have included training results with the following nonlinear activation functions: tanh/sigmoid/ReLU/leaky ReLU. Please refer to the updated Fig. 3c in our Result section for the result of the comparison.
>
> The result shows that our NMDA function with alpha 10 gives the best reference memory performance. We would appreciate any suggestions for other nonlinear functions used in hippocampal-entorhinal models.
>
> **2.2 comment**
> > *This paper claims that NMDAR nonlinearity is needed for long-term memory when it only shows the effect of α on reference memory. What about other types of long-term memory, such as episodic memory?*
>
> We appreciate this feedback and apologize for not definining the scope. While there could be numerous types of long-term memory, such as episodic memory, semantic memory, and procedural memory [1, 2], our focus on long-term memory is considerably limited.
>
> We will update the mansucript by using the terms “short-term working memory (STWM)” and “long-term reference memory(LTRM)” to avoid any confusion. In our 2D spatial navigation task, our focus on long-term memory only addresses the reference memory, and may not be able to address episodic memory, due to the lack of concept of “episode” in this paper.
>
> We have added a new subsection titled “Detailed description of task design and definition of short-term working memory and long-term reference memory” in Appendix A.3.
>
> Our task is based on a widely employed neuroscience experiment for spatial working memory and reference memory [3, 4]. Errors in working memory are measured by within-trial error, whereas errors in reference memory are measured by across-trial error. The training phase and the test phase alternate at each trial. In the test phase, the unvisited place prediction error and visited place prediction error for the familiar map and the novel map, respectively, are measured. The memory of a relatively recent experience can be defined as *short-term working memory* (STWM), and the memory of a relatively old experience can be defined as *long-term reference memory* (LTRM). Within trial visited place prediction measures relatively short-term experience for our task. On the other hand, across trial unvisited place prediction task in the familiar map measures the relatively long-term experience. Measuring unvisited place prediction error in the novel map will establish a baseline of chance-level accuracy; above this baseline, the formation of long-term memory can be observed (Fig. 7).
>
> We have added a new figure with the title, “Detailed task design of working and reference memory evaluation” in Fig. 7
>
> **2.3 comment**
> > *What are the mathematical implications of replacing GELU with NMDA-α? It would strengthen the paper to see a comparison between the GELU-based Transformer and NMDA-α based Transformer in terms of behavioral performance and place cell representations.*
>
> Thank you for the thoughtful comment. To strengthen our paper, we included 1) mathematical implications of replacing GELU with NMDA-α, 2) compared behavioral performance (NMDA-α vs GELU, ReLU, Sigmoid, Tanh, LeakyReLU), and 3) compared place cell representations (NMDA-α vs GELU, ReLU, Sigmoid, Tanh, LeakyReLU).
>
>
> 1) Mathematical implications of replacing GELU with NMDA-α
> In the subsection of the Appendix entitled  “NMDAR-inspired nonlinear activation function”, we derive the generalized NMDA-inspired activation function, NMDAα,β(x), from the NMDAR nonlinearity and compared it with GELU. _Please see our Appendix Section A.1 and A.2 for the NMDAR-inspired activation function._
>
> 2) Compared behavioral performance (NMDA-α vs GELU, ReLU, Sigmoid, Tanh, LeakyReLU)
> We also compared the behavioral performance with GELU and found that NMDAα=10 performs better than any other nonlinear activation function. _Please see our revised main figure 3._
>
> 3) Compared place cell representations (NMDA-α vs GELU, ReLU, Sigmoid, Tanh, LeakyReLU).
> We show the place cell representation in other activation functions and show that NMDAα=10 has the highest performance as well as place cell representations compared to other activation functions. _Please see our revised main figure 4._

---

> ### Author Response · Authors · 2022-11-27
> **Dear reviewer oq3h**
>
> We thank you for your thoughtful reviews and your responses. We have improved our manuscript based on your comments and it would be great if you give us further feedback on our revised version of the manuscript.

---

### Official Review · Reviewer_91tN · 2022-10-25

**Confidence:** 3
**Correctness:** 3
**Technical Novelty And Significance:** 3
**Empirical Novelty And Significance:** 3
**Recommendation:** 5

**Clarity, Quality, Novelty And Reproducibility:**

The key idea, method and findings of this paper are original.

The quality of empirical study is good.

But the clarity of the presentation can be improved. The main text should provide more details about the transformer architecture used in this paper. Some discussions on its biological plausibility may also help.

**Strength And Weaknesses:**

Strengths:

(1) The analogy between the transformer model for prediction and the hippocampus in the brain is interesting, although this analogy has been explored in a recent paper.

(2) The focus on the feedforward block of the transformer model seems novel, and the connection between the GeLU non-linearity and the NMDA receptor (NMDAR) is novel.

(3) The idea and method in this paper is simple and interesting.

Weaknesses:

(1) The paper is entirely empirical. There is no theoretical or mathematical analysis. The empirical similarities between transformer and hippocampus are noteworthy, but some theoretical understanding can greatly improve the paper.

(2) The focus on transformer architecture is understandable given its empirical successes and its popularity, but it may be worthwhile to explore simpler models that may have similar behaviors.


**Summary Of The Paper:**

This paper applies the transformer model to spatial navigation problem in a grid world with labeled grid positions. The task is to predict the label of the next position that is either visited or unvisited. The paper connects this task to working memory and reference memory, as well as place cells. The finding is that NMDAR-like nonlinearity in the feedforward block of the transformer model is important for reference memory and neurons behave like place cells.

**Summary Of The Review:**

The paper compares transformer and hippocampus in navigation tasks, and the findings are interesting. However, the paper is too empirical with no theoretical investigation.

---

> ### Author Response · Authors · 2022-11-13
> **Author response to reviewer 91tN [2/2]**
>
>
>  In neuroscience, the transfer of short-term memory into a long-term memory system is called memory consolidation [6]. Previous animal experiments revealed that the hippocampal CA1 plays an essential role in memory consolidation [7, 8]. In hippocampal CA1, the postsynaptic NMDA receptor mediates synaptic plasticity and the selective perturbation of these receptors leads to impairment in long-term memory formation [8, 9]. Later investigations regarding Mg$^{2+}$-gating of NMDA receptors were shown to modulate long-term memory formation [10, 11]. These lines of evidence suggest that nonlinear dynamics of NMDA receptors in the CA1 are critical for the consolidation of short-term memory into long-term memory. In our work, based on the previous link between the hippocampus and the transformer, we hypothesized transformer can be seen as a memory consolidation model. Given the resemblance of the GELU nonlinear activation function and CA1 NMDAR nonlinear IV curve, we assumed that the GELU activation function serves as a key component that links short-term working memory and long-term reference memory. Based on this assumption, our experimental results show that when the activation function is completely linear (corresponding to no Mg$^{2+}$), the long-term reference memory formation is impaired. In contrast, increasing the $\alpha$ (which corresponds to an increase in Mg$^{2+}$ level) has led our model to have the best performance in long-term reference memory formation compared to other existing common activation functions (RELU, GELU, LRELU, Sigmoid, Tanh). Based on these similarities between hippocampal memory consolidation and our results, we propose a transformer as an effective memory consolidation model.
>
>  In addition to the performance gain in long-term memory formation with NMDA$_\alpha$, we found that modulating the $\alpha$ affects the emergences of place cells in the feed-forward layer, and shows a significant correlation between place cell score and long-term reference memory formation. Our results are in line with previous biological findings that perturbation of CA1 NMDARs lead to impairment in both place cell representation and long-term memory formation [7, 9, 12, 13]. These similarities together support the idea that place cells are the neural correlates of long-term spatial memories. Altogether, our results suggest the interesting possibility that the nonlinear IV curve of NMDAR in the hippocampal CA1 is a neural substrate of nonlinear activation function in the brain.
>
>
> **References**
> [1] Whittington, James CR, et al. "The Tolman-Eichenbaum machine: unifying space and relational memory through generalization in the hippocampal formation." Cell 183.5 (2020): 1249-1263.
> [2] Whittington, James CR, Joseph Warren, and Tim EJ Behrens. "Relating transformers to models and neural representations of the hippocampal formation." International Conference on Learning Representations (2022).
> [3] Cueva, Christopher J., and Xue-Xin Wei. "Emergence of grid-like representations by training recurrent neural networks to perform spatial localization." International Conference on Learning Representations. 2018.
> [4] Banino, Andrea, et al. "Vector-based navigation using grid-like representations in artificial agents." Nature 557.7705 (2018): 429-433.
> [5] Atkinson, Richard C., and Richard M. Shiffrin. "Human memory: A proposed system and its control processes." Psychology of learning and motivation. Vol. 2. Academic Press, 1968. 89-195.
> [6] McGaugh, James L. "Memory--a century of consolidation." Science 287.5451 (2000): 248-251.
> [7] Shimizu, Eiji, et al. "NMDA receptor-dependent synaptic reinforcement as a crucial process for memory consolidation." Science 290.5494 (2000): 1170-1174.
> [8] Remondes, Miguel, and Erin M. Schuman. "Role for a cortical input to hippocampal area CA1 in the consolidation of a long-term memory." Nature 431.7009 (2004): 699-703.
> [9] Tsien, Joe Z., Patricio T. Huerta, and Susumu Tonegawa. "The essential role of hippocampal CA1 NMDA receptor–dependent synaptic plasticity in spatial memory." Cell 87.7 (1996): 1327-1338.
> [10] Slutsky, Inna, et al. "Enhancement of learning and memory by elevating brain magnesium." Neuron 65.2 (2010): 165-177.
> [11] Miyashita, Tomoyuki, et al. "Mg2+ block of Drosophila NMDA receptors is required for long-term memory formation and CREB-dependent gene expression." Neuron 74.5 (2012): 887-898.
> [12] McHugh, Thomas J., et al. "Impaired hippocampal representation of space in CA1-specific NMDAR1 knockout mice." Cell 87.7 (1996): 1339-1349.
> [13] Kentros, Clifford, et al. "Abolition of long-term stability of new hippocampal place cell maps by NMDA receptor blockade." Science 280.5372 (1998): 2121-2126.

---

> ### Author Response · Authors · 2022-11-13
> **Author response to reviewer 91tN [1/2]**
>
> **Overall Response**
>
> The authors thank the reviewers for insightful feedback that helped improve our work. Please find our response to each comment below. We have also made substantial changes to the manuscript. We will be happy to follow up on additional feedback.
>
> **1.1 comment**
> > *The paper is entirely empirical. There is no theoretical or mathematical analysis. The empirical similarities between transformer and hippocampus are noteworthy, but some theoretical understanding can greatly improve the paper.*
>
> Thank you for this important feedback. We now have included the theoretical background and mathematical analysis regarding NMDAR nonlinearity and NMDAR-inspired activation function, as well as citations to the relevant literature. We incorrectly assumed that including this content would shift the focus of our study, but we are happy to follow the reviewer's suggestions and expand the literature.
> While previous work has provided theoretical insights into the specific shape of NMDAR-inspired activation, we are the first to connect this concept to the transformer structure along with empirical evidence. Our grid space navigation task also brings insights into the role of working memory and reference memory.
>
> Please see our Appendix Section A.1 and A.2 for the NMDAR-inspired activation function and A.3 for the task description.
>
> **1.2 comment**
> > *The focus on transformer architecture is understandable given its empirical successes and its popularity, but it may be worthwhile to explore simpler models that may have similar behaviors.*
>
> We appreciate this feedback and the opportunity to clarify our problem better.  We were motivated by recent research examining the relationship between the hippocampus and TEM-transformer (TEM-t) [2] and the emergence of place cells during spatial navigation tasks.
>
> As suggested by the reviewer, other studies have employed simpler models. For example, the RNN-based model is considered in Cueva et al. [3], and LSTM is considered in Banino et al. [4]. Both models explicitly train the agent’s coordinates (i.e., x, y coordinate or step distance and head angle), which is not a desired property in our task.
>
> To examine the emergence of place and grid cells during spatial navigation learning, we chose TEM [1] and TEM-t [2] models that only train with relational information of sensory observation and action sequences. We recognize that we did not adequately explain this rationale in the paper. We will add reviews of simper models in the literature and describe the design choice more clearly.
>
> **1.3 comment**
> >*The main text should provide more details about the transformer architecture used in this paper.*
>
> R: Thank you for the suggestion. We have added a new section (called "Transformer") following the Introduction section.
>
> **1.4 comment**
> >*Some discussions on its biological plausibility may also help.*
>
> We have added a new subsection (called “Transformer as a memory consolidation model and its biological plausibility”) in the Appendix as follows:
>
> In this work, we investigated the biologically inspired NMDA$_\alpha$ activation function in the transformer’s feed-forward layer and its role in memory formation and place cell representation. We show that modulating $\alpha$ corresponds to a change in extracellular [Mg$^{2+}$], by deriving the nonlinear activation function from the real NMDAR nonlinear IV curve. We show the reconstructed real nonlinear IV curve in Fig. 1a (right top).
>
>  In our work, we showed the modulation of $\alpha$ selectively affects the long-term reference memory formation (across trial unvisited place prediction) while leaving the short-term working memory formation (within trial visited place prediction) intact. This result implies that the short-term working memory and long-term reference memory are stored in physically distinct structures; the self-attention layer and feed-forward layer respectively. In psychology, the idea of a multi-store model regarding short-term memory and long-term memory was historically suggested by Atkinson and Shiffrin (1968) [5]. In their model, sensory inputs are stored via attention in short-term memory systems and some of them are transferred to a long-term memory system while others disappear shortly after.

---

> ### Author Response · Authors · 2022-11-27
> **Dear reviewer 91tN**
>
> We thank you for your thoughtful reviews and your responses. We have improved our manuscript based on your comments and it would be great if you give us further feedback on our revised version of the manuscript.

---

### Comment · Area_Chair_5iwA · 2022-11-18
**Responses**

Dear Reviewers,

Thank you for you reviews and your responses. Do you have any further comments for the authors and has it changed you opinion and score of the paper?

Kind regards
AC

---

### Decision · Program_Chairs · 2023-01-20

**Decision:**

Reject

**Justification For Why Not Higher Score:**

While the paper is interesting, it is not strong enough for iclr (and not the best venue for it).

**Justification For Why Not Lower Score:**

N/A

**Metareview: Summary, Strengths And Weaknesses:**

The paper makes interesting connections between the NMDA receptor in the Hippocampus and gelu nonlinearity in transformers. In response to reviewer's comments, the authors toned down the statement of "transformer needs ...". Indeed Relu nonlinearity works pretty well, just slightly worse than gelu (and NMDA with the right setting), consistent with researchers switching from the former to latter to improve the performance.
Overall the reviewers find this paper interesting, but not good enough and not the right venue for iclr, but likely a good workshop paper or a more neuroscience oriented venue.